# ON THE OUT-OF-DISTRIBUTION GENERALIZATION OF SELF-SUPERVISED LEARNING

## ABSTRACT

In this paper, we focus on the out-of-distribution (OOD) generalization of self-supervised learning (SSL). By analyzing the mini-batch construction during the SSL training phase, we first give one plausible explanation for SSL having OOD generalization. Then, from the perspective of data generation and causal inference, we analyze and conclude that SSL learns spurious correlations during the training process, which leads to a reduction in OOD generalization. To address this issue, we propose a post-intervention distribution (PID) grounded in the Structural Causal Model. PID offers a scenario where the spurious variable and label variable is mutually independent. Besides, we demonstrate that if each mini-batch during SSL training satisfies PID, the resulting SSL model can achieve optimal worst-case OOD performance. This motivates us to develop a batch sampling strategy that enforces PID constraints through the learning of a latent variable model. Through theoretical analysis, we demonstrate the identifiability of the latent variable model and validate the effectiveness of the proposed sampling strategy. Experiments conducted on various downstream OOD tasks demonstrate the effectiveness of the proposed sampling strategy.

## 1 INTRODUCTION

Self-supervised learning (SSL) has emerged as a powerful paradigm for training machine learning models without relying on labeled data. SSL models aim to generate general-purpose representations and are typically used as pre-trained weights to effectively initialize downstream tasks. They have demonstrated significant progress in computer vision, achieving competitive or superior performance on various downstream tasks compared to supervised learning approaches (Chen et al., 2020; Grill et al., 2020a; Zbontar et al., 2021; He et al., 2022; Tong et al., 2022). However, despite their superior performance, SSL models face significant challenges in generalizing to out-of-distribution (OOD) data. Understanding and improving the OOD generalization capabilities of SSL is crucial for deploying these models in real-world scenarios where the data distribution can shift over time.

To investigate the OOD generalization properties of SSL, we propose examining the batch construction process during training. SSL methods are generally categorized into two main types: discrimination-based SSL (D-SSL) (Chen et al., 2020; Grill et al., 2020a) and generation-based SSL (G-SSL) (He et al., 2022; Tong et al., 2022). The core principle of D-SSL is augmentation invariance, ensuring that the feature representations of two different augmentations of the same sample are similar. In contrast, G-SSL focuses on the mask and reconstruction principle, where a portion of a sample is masked and then reconstructed using an encoder-decoder structure. Leveraging these principles, augmented samples derived from the same original sample, as well as samples before and after masking, can be considered anchor-related pairs. During SSL training, each pair is treated as a distinct class, effectively framing each mini-batch as a multi-class learning task. Consequently, the SSL training process can be perceived as learning a distribution over tasks based on discrete training tasks, enabling the trained SSL model to generalize to new, unseen tasks, thus demonstrating its OOD generalization capability. However, machine learning is prone to learning spurious correlations that vary between classes and environments (Wang et al., 2023a; 2022). Therefore, although SSL is highly effective in OOD generalization, from a multi-task perspective, different mini-batches in the SSL training process can be considered as different tasks or environments. Consequently, it may still face the challenge of mitigating spurious correlations."

Building upon the analysis presented in Section 3, we examine the aforementioned challenge from the perspectives of data generation and causal inference. First, we conclude that the similarity or reconstruction between samples within a pair is affected by several unobservable factors, such as background or texture information independent of the foreground. We also find that the spurious correlation between the anchor and the unobservable variable can vary with the tasks, making it difficult to eliminate it using the unified causal criterion proposed by (Pearl et al.; Pearl, 2009). Furthermore, we demonstrate that, under these circumstances, the SSL model learns to measure similarity or reconstruct using spurious causal factors. This reliance leads to a lack of discriminability within each mini-batch task, preventing the SSL model from effectively learning the true task distribution and consequently resulting in diminished OOD generalization. To address this issue, we define a new distribution called the post-intervention distribution (PID), characterized by mutual independence between the unobservable variable and the anchor. We demonstrate that when the task distribution adheres to PID, the SSL model trained under this condition achieves the lowest worst-case risk, thereby attaining optimal worst-case OOD performance. This insight motivates us to design a new mini-batch sampling strategy that ensures the resulting mini-batches satisfy PID constraints, thereby enhancing the OOD generalization capability of SSL.

Based on the above analysis and discussion, we propose a novel mini-batch sampling strategy consisting of two stages. In the first stage, we aim to learn a latent variable model to capture the correlations between different variables, i.e., conditional distributions. We prove the identifiability and uniqueness of the resulting latent variable model under a given equivalence relation. In the second stage, we propose a sufficient condition to obtain the balancing score. Using this, we obtain the mini-batch samples through balancing score matching. We also provide a theoretical guarantee that the mini-batches obtained by the proposed sampling strategy approximately satisfy the PID.

In summary, we make the following contributions: **1**) Analysis of SSL Batch Construction: We provide a detailed analysis of how mini-batch construction in SSL influences OOD generalization; **2**) Causal Framework for SSL: We introduce a causal framework to understand and mitigate the impact of spurious correlations on SSL models; **3**) PID-Based Sampling Strategy: We propose a theoretically grounded mini-batch sampling strategy that ensures the generated batches conform to PID, improving OOD performance; **4**) Empirical Validation: We validate our approach through extensive experiments, demonstrating significant improvements in OOD generalization across multiple tasks.

## 2 Revisiting SSL from a Pairwise Perspective

During the training phase, the training data is structured into mini-batches, with each mini-batch denoted as $X_{tr} = \{x_i\}_{i=1}^N$, where $x_i$ represents the $i$-th sample and $N$ is the total number of samples. In D-SSL methods such as SimCLR (Chen et al., 2020), BYOL (Grill et al., 2020a), and Barlow Twins (Zbontar et al., 2021), each sample in $X_{tr}$ undergoes stochastic data augmentation to generate two augmented views, e.g., for $x_i \in X_{tr}$, the augmented samples can be represented as $x_i^1$ and $x_i^2$. For G-SSL methods, like MAE (He et al., 2022) and VideoMAE (Tong et al., 2022), $x_i$ is first divided into multiple small blocks, with some blocks masked, and the remaining blocks reassembled into a new sample, denoted as $x_i^1$. The original sample is then referred to as $x_i^2$. Thus, the augmented dataset in SSL (whether D-SSL or G-SSL) is represented as $X_{tr}^{aug} = \{x_i^1, x_i^2\}_{i=1}^N$. The pair $\{x_i^1, x_i^2\}$ forms the $i$-th pair, and SSL aims to learn a feature extractor $f$ from these pairs.

The objective of D-SSL methods typically consists of two components: alignment and regularization (Wang & Isola, 2020; Chen et al., 2021a). The alignment part is to maximize the similarity between samples that share the same pair in the embedding space, and the regularization part aims to constrain the learning behavior via inductive bias, e.g., SimCLR (Chen et al., 2020) constrains the feature distribution to satisfy a uniform distribution. Meanwhile, G-SSL methods (He et al., 2022) can be regarded as implementing alignment of samples within a pair based on an encoding-decoding structure, by inputting sample $x_i^1$ into this structure to generate a sample, and making it as consistent as possible with sample $x_i^2$. It is noteworthy that "alignment" in D-SSL is often implemented based on anchor points, that is, viewing one sample in a pair as an anchor, the training process of such SSL methods can be seen as gradually pulling the other sample in this pair towards the anchor. The concept of anchor is also applicable to G-SSL, where $x_i^2$ is viewed as the anchor, and thus the training process of such SSL methods can be viewed as gradually constraining $x_i^1$ to approach $x_i^2$.

Based on the above discussion, when we consider the anchor as the label or the center of clustering, each mini-batch in the SSL training phase thus can be viewed as a multi-class classification task. Specifically, $X_{tr}^{aug} = \{x_i^+, x_i^{\text{anchor}}\}_{i=1}^{2N}$ consists of data from $N$ categories, where $x_i^+$ is the positive sample of the $i$-th category whose clustering center is $x_i^{\text{anchor}}$. Furthermore, the variability of data across mini-batches implies that each mini-batch corresponds to a distinct training task or domain.

# 3 MOTIVATION AND CAUSAL ANALYSIS

In this section, we first offer a plausible explanation for the OOD generalization capability of SSL models from a task distribution perspective. Next, based on data generation principle and causal inference, we demonstrate that SSL methods may measure similarity or reconstruction using spurious correlations between pairs, which reduces their OOD generalization performance. Finally, through theoretical analysis, we present that even in the case of spurious associations, we can further improve the OOD generalization of SSL by constraining the data distribution.

## 3.1 FORMATION OF THE PROBLEM: CAUSAL PERSPECTIVE

According to Section 2, different mini-batches correspond to distinct classification tasks. Therefore, the training process of SSL can be described as follows: given a distribution over tasks and a data distribution for each task (refer to **Appendix** E for more details), the SSL model is learned based on various training tasks and their corresponding data. The performance of the SSL model is then evaluated on test tasks that are disjoint from the training tasks. This learning paradigm involves estimating the true task distribution from discrete training tasks (refer to **Appendix** E for more details), enabling the SSL model to generalize to new, unseen tasks (i.e., test tasks). This also explains well why the SSL model exhibits good performance in transfer tasks (Chen et al., 2020; Grill et al., 2020a; Zbontar et al., 2021), i.e., it has good OOD generalization. However, machine learning models are prone to learning spurious correlations during the training phase (Wang et al., 2023a; 2022). For example, compared to the foreground features of input data, researchers have found that machine learning models tend to rely on the superficial texture information or background information of the data for decision-making (Geirhos et al., 2018; Qiang et al., 2022; Xu et al., 2020). Therefore, although the SSL model has been effective in OOD generalization, we find that it still faces the challenge of spurious correlations.

We further analyze the above challenge from the perspective of data generation and causal inference. Without loss of generality, for each pair in the SSL training process, we denote the anchor as $x^{\text{label}}$ and the other sample as $x^+$. Based on (Zimmermann et al., 2021; Von Kügelgen et al., 2021), $x^+$ can be regarded as caused by anchor $x^{\text{label}}$, an unobserved latent variable $s \in \mathbb{R}^n$ and an independent noise variable $\epsilon$ with the following formulation:

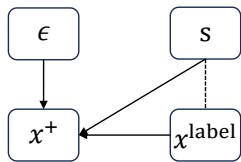

Figure 1: The SCM for Equation (1).

$$x^+ = F(s, x^{\text{label}}) + \epsilon, \tag{1}$$

where $F$ is a reversible injective function. From a causal perspective, Equation (1) can be reformulated as the Structural Causal Model (SCM) shown in Figure 1. The solid arrow indicates that there is a direct causal relationship between the two variables, e.g., $x^{\text{label}} \rightarrow x^+$ states that $x^{\text{label}}$ is the direct cause of obtaining $x^+$. The dotted line indicates that the relationship between the variables is not clear and varies with different environments. Notably, this paper focuses exclusively on scenarios where the semantic information within $x^+$ is related only to $x^{\text{label}}$, that is, $s$ does not contain any causal semantics related to the task. Next, we examine two examples illustrated in Figure 2. In Figure 2 (a), $s$ represents the assigned color, for example, the color of numbers varies by category, as in the ColoredMNIST dataset (Arjovsky et al., 2019). Here, $e_{\text{id}}$ denotes the class index. Consequently, within a mini-batch during training, samples from different classes may have a different texture color. In Figure 2 (b), $s$ indicates assigned stylistic attributes, e.g., sketches, cartoon styles, or photographs, and $e_{\text{id}}$ denotes the batch index. This scenario commonly occurs in multi-view or domain generalization contexts, like the tasks in the PACS dataset (Li et al., 2017). Therefore, during training, different batches may exhibit different styles, with samples under each style possessing

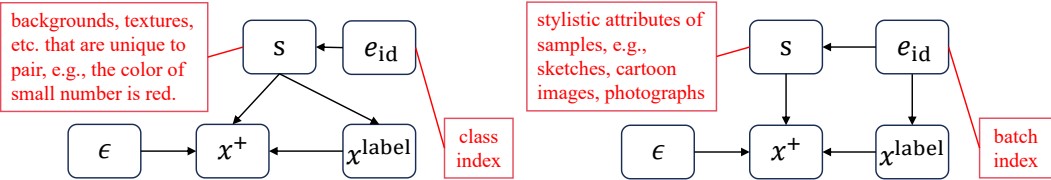

(a) Example task related to ColoredMNIST dataset     (b) Example task related to PACS dataset

Figure 2: Two specific instances illustrate the variability in the causal relationship between $x^{\text{label}}$ and $s$ due to environmental changes. The black squares are variables and the arrows indicate causality.

unique appearance attributes. In both figures, $s$ does not capture the foreground semantics between $x^{\text{label}}$ and $x^+$, and the correlation between $x^{\text{label}}$ and $x^+$ may vary depending on the settings.

Based on Figure 2 (a) and (b), we obtain that the causal relationship between $x^{\text{label}}$ and $s$ changes with unknown environmental variations, making it difficult to eliminate based on a unified causal criterion proposed in (Pearl et al.). From Figure 2 (a), due to the existence of path $x^{\text{label}} \cdots \cdots s \rightarrow x^+$, the following proposition states that the correlation between $x^{\text{label}}$ and $x^+$ is influenced by $s$.

**Proposition 3.1** *Revisiting SSL from a pairwise perspective and assuming that the two samples in each pair satisfy Equation* (1)*, we can obtain that the learned SSL model will use non-causal factor, i.e., the unobserved latent variable $s$, to measure the similarity or reconstruct in a pair.*

Detailed proof of **Proposition** 3.1 is provided in **Appendix** A.1. Notably, when SSL models measure the similarity or reconstruct between paired elements using non-causal factors, the extracted representations may incorporate semantics irrelevant to the task. From the pairwise perspective, this may result in SSL failing to effectively learn each specific task, thereby hindering the modeling of the task distribution and ultimately reducing the OOD generalization ability of SSL.

### 3.2 Motivation: Post-Intervention Distribution

As shown in Figure 2, regardless of the correlation between $s$ and $x^{\text{label}}$, the generation mechanism of $x^+$ is invariant. Becuase SCMs can also be considered as a joint probability distribution, thus, we use the following distribution set to represent the joint probability distribution related to Figure 1:

$$\mathcal{D} = \left\{ p(x^+, x^{\text{label}}, s) = p(x^+|x^{\text{label}}, s)p(x^{\text{label}})p(s|x^{\text{label}}) \Big| p(x^{\text{label}}), p(s|x^{\text{label}}) > 0 \right\}. \quad (2)$$

Instead of exploring what the specific structure of $x^{\text{label}} \cdots \cdots s \rightarrow x^+$, we propose to consider using Post-Intervention Distribution (PID) to model $p(x^+, x^{\text{label}}, s)$, which can be defined as:

**Definition 3.2** *If $p(x^+, x^{\text{label}}, s) = p(x^+|x^{\text{label}}, s)p(x^{\text{label}})p(s)$, then $p(x^+, x^{\text{label}}, s)$ is defined as PID. In other words, $x^{\text{label}}$ and $s$ are independent in PID.*

We use $p^{\text{PI}}$ to denote distributions belonging to the PID family. As we can see, $p(x^+|x^{\text{label}}, s)$ is both a component of $p^{\text{PI}}(x^+, x^{\text{label}}, s)$ and a result of the unchanged causal mechanism $s \rightarrow x^+ \leftarrow x^{\text{label}}$ in Figure 1. Then, the corresponding SCM of $p^{\text{PI}}(x^+, x^{\text{label}}, s)$ is shown as Figure 3. In this new distribution, because there are no paths between $s$ and $x^{\text{label}}$, we can obtain that $x^+$ and $x^{\text{label}}$ are only correlated through the stable causal relation

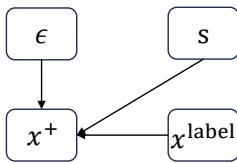

Figure 3: The SCM for $p^{\text{PI}}(x^+, x^{\text{label}}, s)$.

$x^+ \leftarrow x^{\text{label}}$. Then, from a probabilistic perspective, what we argue is that compared to SSL models trained on batches satisfying other distribution constraints in $\mathcal{D}$, SSL models trained on batches that meet the PID distribution constraint have the lowest worst-case risk. To support this statement, we build upon (Pearl, 2009) by introducing an assumption regarding the invertibility of functions:

**Assumption 3.3** *There exist functions $F_{x^{\text{label}}}$ , $F_s$ and noise variables $\epsilon_{x^{\text{label}}}$ , $\epsilon_s$ , such that $(x^{\text{label}}, s) = F^{-1}(x^+ - \epsilon) = (F_{x^{\text{label}}}(x^+ - \epsilon_{x^{\text{label}}}), F_s(x^+ - \epsilon_s))$, and $\varepsilon_{x^{\text{label}}} \perp\!\!\!\perp_{\text{PI}} \epsilon_s$.*

**Assumption** 3.3 implies that $x^{\text{label}} \perp\!\!\!\perp_{\text{PI}} s | x^{+}$, and the intuitive explanation of **Assumption** 3.3 can be found in **Appendix** F. Based on Section 2 and Section 3.1, both D-SSL and G-SSL share a common learning objective: aligning the positive sample in a pair with its corresponding anchor. Thus, the learning objectives of D-SSL and G-SSL can be unified as maximizing $p_f(x^{\text{label}} | x^{+})$. The difference lies in how they achieve $p_f(x^{\text{label}} | x^{+})$. For example, the training data is first projected to the feature space by $f$, then SimCLR uses a contrastive loss to achieve $p_f(x^{\text{label}} | x^{+})$, while MAE employs the $L_2$-norm achieve $p_f(x^{\text{label}} | x^{+})$. We can then obtain the following conclusion:

**Theorem 3.4** *From a Bayesian perspective, the alignment part of the SSL learning objective, e.g., constrain samples under the same pair to be similar in the feature space, can be expressed as* $\max p_f(x^{\text{label}} | x^{+})$. *Given $f$, the risk on a batch with $e \in \mathcal{D}$ as the distributional constraint can be presented as:* $\mathcal{L}^e(f) = \mathbb{E}_{p^e(x^{+}, x^{\text{label}})} - \log p_f(x^{\text{label}} | x^{+})$, *where $p^e(x^{+}, x^{\text{label}})$ denotes the joint distribution. Under **Assumption 3.3**, when $f^* = \arg\min \mathcal{L}^e(f)$, s.t. $e \in \text{PID}$, we have $f^*$ is the minimax optimal across all elements in $\mathcal{D}$, e.g., $f^* = \arg_f \min \max_{e \in \mathcal{D}} \mathcal{L}^e(p_f(x^{\text{label}} | x^{+}))$.*

Detailed proof of **Theorem** 3.4 is provided in **Appendix** A.2. **Theorem** 3.4 implies that when $\mathcal{D}$ is sufficiently large and diverse, no other $f$ obtained from training on any distribution can achieve better worst-case OOD performance than the PID. Notably, transferring Figure 1 to Figure 3 is similar to backdoor adjustment in causal inference (Pearl et al.). However, from backdoor adjustment pespective, it is straightforward to explain why PID can improve the OOD performance of D-SSL: during the learning of each task, PID eliminates the influence of background semantic confounding (Qiang et al., 2022). However, for G-SSL, regardless of the relationship between $s$ and $x^{\text{label}}$, G-SSL inherently requires encoding background semantics. Thus, explaining the improvement of OOD performance of G-SSL from the backdoor adjustment perspective is incorrect. Therefore, **Theorem** 3.4 is provided to explain why PID can improve the OOD performance of both D-SSL and G-SSL simultaneously. Also, an intuitive explanation of **Theorem** 3.4 is shown in **Appendix** F. Moreover, **Theorem** 3.4 motivates us to design a new mini-batch sampling strategy to ensure that the resulting mini-batches satisfy PID, thereby improving the OOD generalization of SSL.

# 4 THE PROPOSED METHOD

In this section, we present the proposed method which consists of two stages. In the first stage, we use a latent variable model, e.g., variational autoencoder (VAE) (Kingma & Welling, 2013a), to learn the underlying distribution $p(x^{+}, x^{\text{label}}, s)$ for each batch task. In the second stage, we use the learned distribution to obtain a sampling strategy that can create a PID based on training data.

## 4.1 LEARNING LATENT VARIABLE MODEL

As shown in Equation (2), to learn the underlying joint distribution $p(x^{+}, x^{\text{label}}, s)$ for each batch task, we need to know $p(x^{+} | x^{\text{label}}, s), p(x^{\text{label}}), p(s | x^{\text{label}})$ in each batch task. Because that $p(x^{+} | x^{\text{label}}, s)$ is the unchanged causal mechanism, so we can use a unified f to model $p(x^{+} | x^{\text{label}}, s)$ in all tasks. Based on the discussion in Section 2, we obtain that $x^{\text{label}}$ is regarded as the label. So, $p(x^{\text{label}})$ can be regarded as the label distribution, and we can represent it with the same uniform distribution in all tasks. Based on the mean-field approximation (Blei et al., 2017; Sriperumbudur et al., 2013) which can be expressed as a closed form of the true prior, we obtain that when the causal relationship between the latent covariate and the label changes with the tasks, an exponential family distribution has the ability to model the conditional distribution $p(s | x^{\text{label}})$, thus, we have the following assumption for each batch task:

**Assumption 4.1** *Denote the mini-batch task index as $e$, the correlation between $x^{\text{label}}$ and $s$ in the data distribution $p^e(x^{+}, x^{\text{label}}, s)$ of a task is characterized by:*

$$p_{\text{T}, \lambda^e}^e(s | x^{\text{label}}) = \prod_{i=1}^{n} \frac{Q_i(s_i)}{K_i^e(x^{\text{label}})} \exp[\sum_{j=1}^{k} T_{ij}(s_i) \lambda_{ij}^e(x^{\text{label}})], \tag{3}$$

*where $n$ is the dimension of the latent variable $s$, $k$ is the dimension of each sufficient statistic, $s_i$ is the $i$-th element of $s$, $\text{Q} = [Q_i]: s \to \mathbb{R}^n$ is the base measure, $\text{T} = [T_{ij}]: s \to \mathbb{R}^{nk}$ is the sufficient statistics, $\text{K}^e = [K_i^e]: x^{\text{label}} \to \mathbb{R}^n$ is the normalizing constraint, and $\lambda^e = [\lambda_{ij}^e]: x^{\text{label}} \to \mathbb{R}^{nk}$.*

Note that $k$, Q, and T are determined by the type of chosen exponential family distribution and thus independent of $e$, this guides us to constrain all batch tasks to share these parameters during the training phase. For ease of calculation, we set $Q_i(\cdot) = \exp(\cdot / - 2)$ and $K^e$ as the feature normalization operator. For $\lambda^e$, since it varies with $e$, we implement it as the output of a network. Specifically, we first average all the data of a batch, then feed it into a learnable network $g$, and output the corresponding $\lambda^e$. For T, we need to guarantee it to be a sufficient statistic, one simple way to implement this is the constant transformation. Considering the identifiability of the parameters, we implement it as $T_{ij}(\cdot) = a_{ij} \times \cdot$, where $A = [a_{ij}]$ is a learnable parameter. Up to this point, we obtain the implementation of $p^e_{T,\lambda^e}(s|x^{\text{label}})$ as $p_{g,A}(s|x^{\text{label}})$. Then, we implement the conditional generative model in each $e \in \mathcal{D}$ with parameters $\theta = (f, g, A)$ as: $p^e_\theta(x^+, s|x^{\text{label}}) = p_f(x^+|s, x^{\text{label}})p_{g,A}(s|x^{\text{label}})$.

Motivated by the VAE, we estimate the above conditional generative model with the following regularized evidence lower bound (ELBO) in each batch distribution $e$:

$$\mathcal{L}^e_{\theta,\phi} = \mathbb{E}_{q_\phi(s|x^+, x^{\text{label}})}[\log p_f(x^+|s, x^{\text{label}})] - \text{KL}(q_\phi(s|x^+, x^{\text{label}}) \| p_{g,A}(s|x^{\text{label}})) - \alpha \sum_{i,j} A_{\cdot,i} \cdot A_{\cdot,j}, \quad (4)$$

where $A_{\cdot,i}$ is the column vector of A, $\text{KL}(\cdot)$ is the KL-divergence, and $\alpha$ is a hyperparameter. As for $q_\phi(s|x^+, x^{\text{label}})$, it is implemented by a learnable network $\phi$ that outputs the mean and variance, and we use reparameterization trick (Kingma & Welling, 2013b) to deal with it during training. The last term of Equation (4) is to constrain the column vector orthogonality of A. The training process of Equation (4) is similar to meta-learning, e.g., Prototype Networks (Snell et al., 2017), because that we construct a series of tasks during the training phase. Thus, from a meta-learning perspective, training with Equation (4) also indicates that the learned $\theta$ can be adaptable for all available tasks.

We further show that we can uniquely recover the model parameter $\theta$ up to an equivalence relation. Specifically, we first give the definition of the equivalence relation based on (Motiian et al., 2017):

**Definition 4.2** $(f, g, A) \sim_W (f', g', A')$, *if and ony if there exists an invertible matrix* $W \in \mathbb{R}^{nk \times nk}$ *and a vector* $b \in \mathbb{R}^{nk}$, *such that* $A(f^{-1}(x)) = WA'(f'^{-1}(x)) + b, \forall x \in X^{aug}_{tr}$.

Then, motivated by (Khemakhem et al., 2020), the identifiability condition of $\theta$ can be presented as:

**Theorem 4.3** *Suppose that* $p^e_\theta(x^+, s|x^{\text{label}}) = p_f(x^+|s, x^{\text{label}})p_{g,A}(s|x^{\text{label}})$ *and the generation process of* $X^+$ *can be represented by the SCM depicted in Figure 1, a sufficient condition for* $\theta = (f, g, A)$ *to be* $\sim_A$*-identifiable is given as: 1) Suppose that* $p_\epsilon(x^+ - f(x^{\text{label}}, s)) = p_f(x^+|x^{\text{label}}, s)$, $\phi_\varepsilon$ *is the characteristic function of* $p_\epsilon(x^+ - f(x^{\text{label}}, s))$, *and the set* $\{x^+|\phi_\varepsilon(x^+) = 0\}$ *has measure zero; 2) The sufficient statistics* T *are differentiable almost everywhere, and* $[T_{ij}]_{1 \leq j \leq k}$ *are linearly independent on any subset of* $X^+$ *with measure greater than zero; 3) There exist* $nk + 1$ *distinct pairs* $(x^{\text{label}}_0, e_0), \cdots, (x^{\text{label}}_n k, e_{nk})$ *such that the* $nk \times nk$ *matrix* $L = (\lambda^{e_1}(x^{\text{label}}_1) - \lambda^{e_0}(x^{\text{label}}_0), \cdots, \lambda^{e_{nk}}(x^{\text{label}}_{nk}) - \lambda^{e_0}(x^{\text{label}}_0))$ *is invertible.*

Detailed proof of **Theorem** 4.3 is provided in **Appendix** A.3. In Equation (4), we constrain the column vector orthogonality of A, this can lead to the linearly independence of elements of T, thus, the second assumption of **Theorem** 4.3 holds. Meanwhile, according to Section 2, we can obtain that each ancestor training sample can be regarded as a class, by combining different classes with each other, we can construct adequate tasks, thus, the third assumption of **Theorem** 4.3 can easily holds. Therefore, based on **Theorem** 4.3, we can obtain that $\theta$ can be uniquely recovered. Moreover, the detailed explanation of the identifiability of spurious variable $s$ is provided in **Appendix** G.

## 4.2 THE PROPOSED MINI-BATCH SAMPLING STRATEGY

As shown in (Rosenbaum & Rubin, 1981), balancing score matching has become a useful tool in the average treatment effect estimation. One of its purposes is to reveal the true causal relationship from the observational data. It is defined as:

**Definition 4.4** *A balancing score* $ba(s)$ *is a function of covariate* $s$ *that satisfies:* $s \perp\!\!\!\perp x^{\text{label}}|ba(s)$.

From (Rosenbaum & Rubin, 1981), we can obtain that many functions can be used as a balancing score, among them, propensity score $p(x^{\text{label}}|s)$ is the coarsest one. Motivated by this, given the batch task with nu pairs, we define the propensity score under the SSL scenario as:

**Definition 4.5** *The propensity score for a batch task in SSL scenario is $mi(s) = [p(x_j^{\text{label}}|s)]_{j=1}^{\text{nu}}$.*

Then, given a function $ba(s)$, we present a sufficient condition that it can be the balancing score:

**Corollary 4.6** *Let $ba(s)$ be a function of $s$, a sufficient condition that $ba(s)$ can be regarded as a balancing score is that there exists a function $\psi$ such that $mi(s) = \psi(ba(s))$.*

The proof of **Corollary** 4.6 can be directly obtained based on **Theorem** 1 and **Theorem** 2 in Rosenbaum & Rubin (1981). We use $ba^e(s)$ to denote the balancing score for a specific batch task $e$ of SSL. Then, the corresponding propensity score can be represented as $mi^e(s) = [p^e(x_j^{\text{label}}|s)]_{j=1}^{\text{nu}}$, which can be derived from $p_{T,\lambda^e}^e(s|x^{\text{label}})$ as defined in Equation (3):

$$p^e(x_j^{\text{label}}|s) = \frac{p_{g,A}(s|x_j^{\text{label}})p^e(x_j^{\text{label}})}{\sum_{j=1}^{\text{nu}} p_{g,A}(s|x_j^{\text{label}})p^e(x_j^{\text{label}})}, \tag{5}$$

where $p^e(x_j^{\text{label}}) = 1/\text{nu}$, because that $p^e(x_j^{\text{label}})$ is defined empirically as a uniform distribution.

Based on **Corollary** 4.6, we set $\psi$ as identical transformation and propose to use the propensity score computed from Equation (5) directly as our balancing score, e.g., $ba(s) = mi^e(s)$. Next, we derive the proposed sampling strategy. When given the training data $X^{tr} = \{x_i^+, x_i^{\text{label}}\}_{i=1}^{\text{mu}}$ with mu pairs, we can obtain $\lambda^e$ of Equation (5) based on the mean of the entire dataset. Then, for each pair, we firstly obtain $s$ based on the learned $q_\phi^e(s|x^+, x^{\text{label}})$ and secondly obtain $ba(s)$ by setting nu = mu in Equation (5). Finally, the proposed sampling strategy is constructed by matching $ba(s)$ of the selected pair with $1 \le a \le N-1$ different pairs that have the same/closest balancing score. The detailed sampling strategy is shown as follows:

---
**Algorithm 1:** The Proposed Mini-Batch Sampling Strategy.

---
**Input:** Training datasets $X^{tr} = \{x_i^+, x_i^{\text{label}}\}_{i=1}^{\text{mu}}$, a balancing score $ba(\cdot)$ inferred from each
  training pair $(x_i^1, x_i^2)$, and a distance metrics $d : ba(\cdot) \times ba(\cdot) \to \mathbb{R}$;
**Output:** A mini-batch of data $D^{\text{PI}}$ consisting of $a + 1$ examples;
$D^{\text{PI}} \leftarrow$ Empty; $i \leftarrow 0$;
**for** $i = 0$ **do**
    Randomly sample a pair $(x_i^+, x_i^{\text{label}})$ from $X_{tr}^{aug}$, add it to $D^{\text{PI}}$;
    Compute balancing score $ba(s_i)$ from $(x_i^+, x_i^{\text{label}})$;
    Set $i \leftarrow i + 1$;
**for** $1 \le i \le a$ **do**
    $j = \arg\min_{x_j^+ \in X_{tr}^{aug} \setminus D^{\text{PI}}} d(ba(s_j), ba(s_i))$;
    Add $(x_j^+, x_j^{\text{label}})$ to $D^{\text{PI}}$;
    Set $i \leftarrow i + 1$.

---

We denote the data distribution obtained from **Algorithm** 1 as $\hat{p}(x^+, x^{\text{label}}, s)$, then we have:

**Theorem 4.7** *If $d(ba(s_j), ba(s_i)) = 0$ in **Algorithm 1**, the obtained mini-batch is regarded as sampling from a PID, e.g., $\hat{p}(x^{\text{label}}|s) = p^{\text{PI}}(x^{\text{label}})$.*

Detailed proof and high-level explanation of **Theorem** 4.7 is provided in **Appendix** A.4 and F. Based on **Theorem** 4.7, if at each step, we achieve perfect matching (i.e., $ba(s_j) = ba(s_i)$), and the obtained mini-batch samples can be regarded as sampled from the PID. However, an exact match of the balancing score is unlikely during the SSL training phase (each pair has only one positive sample), so a larger $a$ can introduce noise. This can be mitigated by selecting a smaller $a$, which increases the dependency between $x^{\text{label}}$ and $s$. Thus, in practice, the choice of $a$ reflects a trade-off between the quality of balancing score matching and the degree of dependency between $x^{\text{label}}$ and $s$.

## 5 EXPERIMENTS

In this section, we first introduce the datasets used in experiments. Next, we evaluate our method on multiple tasks, including unsupervised learning, semi-supervised learning, transfer learning, and

few-shot learning. We introduce the experimental setups in the corresponding sections. Finally, we perform ablation studies. All results reported are the averages of five runs performed on NVIDIA RTX 4090 GPUs. More experiments are shown in **Appendix** C due to space limitations.

## 5.1 BENCHMARK DATASETS

For unsupervised learning, we select ImageNet-100 (Tian et al., 2020) and ImageNet (Deng et al., 2009) for analysis. For semi-supervised learning, we select ImageNet (Deng et al., 2009) for evaluation. For transfer learning, we select PASCAL VOC (Everingham et al., 2010) and COCO (Lin et al., 2014) for analysis. For few-shot learning, we evaluate the proposed method on Omniglot (Lake et al., 2019), miniImageNet (Vinyals et al., 2016), and CIFAR-FS (Bertinetto et al., 2018).

## 5.2 EMPIRICAL ANALYSIS

In this article, we primarily addresses the OOD generalization of SSL. Our experimental design consists of the following steps: First, we validate that the proposed sampling strategy enhances the performance of SSL methods in in-distribution scenarios using unsupervised tasks. Second, we classify OOD tasks by difficulty into semi-supervised tasks, transfer learning tasks, and few-shot learning tasks, and subsequently evaluate the proposed sampling strategy on these tasks. Meanwhile, we also conduct experiments on generative SSL, the evaluation are provided in **Appendix** C.1.

**Experimental setup.** Our proposed sampling strategy can be applied to any D-SSL and G-SSL models. It only changes the mini-batch generation mechanism without affecting the training process or altering the hyperparameter settings. Therefore, the hyperparameter settings for all our experiments are consistent with the methods we are comparing, and we will not elaborate on them here.

**Results on unsupervised learning tasks. Table** 1 shows the top-1 and top-5 linear classification accuracies on ImageNet-100 and ImageNet for unsupervised learning task. We can observe that applying the proposed method achieves stable performance improvement, and significantly outperforms the state-of-the-art (SOTA) methods on all datasets and all the SSL baselines.

**Results on semi-supervised learning tasks. Table** 2 shows the results on ImageNet for semi-supervised learning task. We can observe that no matter 1% or 10% of the labels are available in 1000 epochs, the improvement brought by the proposed methods reaches more than 3% on Top-1 and 2% on Top-5 results. This further demonstrate the effectiveness of the proposed method.

**Results on transfer learning tasks. Table** 3 shows the results on the most commonly used object detection and instance segmentation protocol Chen et al. (2020); Zbontar et al. (2021) for transfer learning. The results shows that introducing the proposed method achieve stable improvements in all the metrics, tasks, and baselines, reaching an average improvement of nearly 3.8%.

**Results on few-shot learning tasks. Table** 4 shows the effect of the proposed sampling strategy on standard few-shot transfer learning tasks. From the results, we can see that compared to the original baselines, introducing our proposed method achieves remarkable performance improvement, achieving more than 5% improvement. These results demonstrate the superiority of the proposed method under data-scarce conditions and further proves its effectiveness.

In summary, from all the experimental results, we can observe that when the SSL methods are trained based on mini-batches generated by our proposed sampling strategy, they all further improve their performance and by at least 2%. This shows that our sampling strategy is effective in further reducing the false correlation information in the distribution of the mini-batch task, which leads to better causal learning and improves the OOD generalization of the SSL model.

## 5.3 ABLATION STUDY

**Influence of the batch size hyperparameter a.** According to **Algorithm** 1, $a$ is the hyperparameter of the proposed sampling strategy, which represent the batch size. As shown in **Theorem** 4.7, we can obtain that a suitable $a$ is important. To explore whether the SSL model is more sensitive to the original batch size or to $a$, we conduct experiments based on ImageNet and BYOL, and the corresponding results are shown in Figure 4. We can observe that the performance of BYOL rapidly deteriorates with batch size. In contrast, the performance of BYOL + Ours remains stable over a

Table 1: The Top-1 and Top-5 classification accuracies of linear classifier on the ImageNet-100 dataset and the Top-1 results for ImageNet dataset with ResNet-50 as feature extractor.

| Method | ImageNet-100 | | ImageNet | |
| --- | --- | --- | --- | --- |
| | Top-1 | Top-5 | 400 Epochs | 1000 Epochs |
| SimCLR (Chen et al., 2020) | 70.15 ± 0.16 | 89.75 ± 0.14 | 69.24 ± 0.21 | 70.45 ± 0.30 |
| MoCo (He et al., 2020) | 72.80 ± 0.12 | 91.64 ± 0.11 | 69.76 ± 0.14 | 71.16 ± 0.23 |
| SimSiam (Chen & He, 2021) | 73.01 ± 0.21 | 92.61 ± 0.27 | 70.86 ± 0.34 | 71.37 ± 0.22 |
| Barlow Twins (Zbontar et al., 2021) | 75.97 ± 0.23 | 92.91 ± 0.19 | 70.22 ± 0.15 | 73.29 ± 0.13 |
| SwAV (Caron et al., 2020) | 75.78 ± 0.16 | 92.86 ± 0.15 | 70.78 ± 0.34 | 75.32 ± 0.11 |
| DINO (Caron et al., 2021) | 75.43 ± 0.18 | 93.32 ± 0.19 | 71.98 ± 0.26 | 73.94 ± 0.29 |
| RELIC v2 (Tomasev et al., 2022) | 75.88 ± 0.15 | 93.52 ± 0.13 | 71.84 ± 0.21 | 72.17 ± 0.20 |
| MEC (Liu et al., 2022a) | 75.38 ± 0.17 | 92.84 ± 0.20 | 72.91 ± 0.27 | 75.07 ± 0.24 |
| VICRegL (Bardes et al., 2022) | 75.96 ± 0.19 | 92.97 ± 0.26 | 72.14 ± 0.20 | 75.07 ± 0.23 |
| SimCLR + Ours | 73.32 ± 0.15 | 91.74 ± 0.18 | 72.24 ± 0.20 | 73.66 ± 0.25 |
| MoCo + Ours | 74.71 ± 0.24 | 93.89 ± 0.17 | 72.04 ± 0.21 | 74.06 ± 0.20 |
| SimSiam + Ours | 75.66 ± 0.18 | 95.02 ± 0.21 | 72.96 ± 0.22 | 73.67 ± 0.17 |
| Barlow Twins + Ours | 77.77 ± 0.18 | 94.99 ± 0.20 | 73.08 ± 0.21 | 75.89 ± 0.17 |
| SwAV + Ours | 76.99 ± 0.11 | 95.03 ± 0.20 | 73.25 ± 0.24 | 77.42 ± 0.21 |
| DINO + Ours | 77.47 ± 0.15 | **96.01** ± 0.17 | 74.21 ± 0.20 | 75.99 ± 0.17 |
| VICRegL + Ours | **78.20** ± 0.14 | 95.07 ± 0.21 | **74.91** ± 0.14 | **77.77** ± 0.21 |

Table 2: The semi-supervised learning accuracies (± 95% confidence interval) on the ImageNet dataset with the ResNet-50 pre-trained on the Imagenet dataset.

| Method | Epochs | 1% | | 10% | |
| --- | --- | --- | --- | --- | --- |
| | | Top-1 | Top-5 | Top-1 | Top-5 |
| MoCo (He et al., 2020) | 200 | 43.8 ± 0.2 | 72.3 ± 0.1 | 61.9 ± 0.1 | 84.6 ± 0.2 |
| BYOL (Grill et al., 2020b) | 200 | 54.8 ± 0.2 | 78.8 ± 0.1 | 68.0 ± 0.2 | 88.5 ± 0.2 |
| BYOL + Ours | 200 | 46.5 ± 0.2 | 74.4 ± 0.2 | 63.6 ± 0.3 | 85.6 ± 0.2 |
| MoCo + Ours | 200 | **57.4** ± 0.2 | **80.1** ± 0.2 | **71.4** ± 0.2 | **90.2** ± 0.1 |
| SimCLR (Chen et al., 2020) | 1000 | 48.3 ± 0.2 | 75.5 ± 0.1 | 65.6 ± 0.1 | 87.8 ± 0.2 |
| MoCo (He et al., 2020) | 1000 | 52.3 ± 0.1 | 77.9 ± 0.2 | 68.4 ± 0.1 | 88.0 ± 0.2 |
| BYOL (Grill et al., 2020b) | 1000 | 56.3 ± 0.2 | 79.6 ± 0.2 | 69.7 ± 0.2 | 89.3 ± 0.1 |
| SimSiam (Chen & He, 2021) | 1000 | 54.9 ± 0.2 | 79.5 ± 0.2 | 68.0 ± 0.1 | 89.0 ± 0.3 |
| Barlow Twins (Zbontar et al., 2021) | 1000 | 55.0 ± 0.1 | 79.2 ± 0.1 | 67.7 ± 0.2 | 89.3 ± 0.2 |
| RELIC v2 (Tomasev et al., 2022) | 1000 | 55.2 ± 0.2 | 80.0 ± 0.1 | 68.0 ± 0.2 | 88.9 ± 0.2 |
| MEC (Liu et al., 2022a) | 1000 | 54.8 ± 0.1 | 79.4 ± 0.2 | 70.0 ± 0.1 | 89.1 ± 0.1 |
| VICRegL (Bardes et al., 2022) | 1000 | 54.9 ± 0.1 | 79.6 ± 0.2 | 67.2 ± 0.1 | 89.4 ± 0.2 |
| SimCLR + Ours | 1000 | 50.8 ± 0.2 | 77.8 ± 0.2 | 67.3 ± 0.1 | 89.9 ± 0.2 |
| MoCo + Ours | 1000 | 53.9 ± 0.2 | 78.9 ± 0.2 | 71.2 ± 0.1 | 89.5 ± 0.1 |
| BYOL + Ours | 1000 | **58.9** ± 0.2 | **81.9** ± 0.2 | **72.1** ± 0.2 | 91.2 ± 0.1 |
| Barlow Twins + Ours | 1000 | 57.6 ± 0.2 | 80.6 ± 0.1 | 68.9 ± 0.2 | **91.8** ± 0.2 |

Table 3: The results of transfer learning on object detection and instance segmentation with C4-backbone as the feature extractor. "AP" is the average precision, "$AP_N$" represents the average precision when the IoU (Intersection and Union Ratio) threshold is $N\%$.

| Method | VOC 07 detection | | | VOC 07+12 detection | | | COCO detection | | | COCO instance segmentation | | |
| --- | --- | --- | --- | --- | --- | --- | --- | --- | --- | --- | --- | --- |
| | $AP_{50}$ | AP | $AP_{75}$ | $AP_{50}$ | AP | $AP_{75}$ | $AP_{50}$ | AP | $AP_{75}$ | $AP_{50}^{mask}$ | $AP^{mask}$ | $AP_{75}^{mask}$ |
| Supervised | 74.4 | 42.4 | 42.7 | 81.3 | 53.5 | 58.8 | 58.2 | 38.2 | 41.2 | 54.7 | 33.3 | 35.2 |
| SimCLR (Chen et al., 2020) | 75.9 | 46.8 | 50.1 | 81.8 | 55.5 | 61.4 | 57.7 | 37.9 | 40.9 | 54.6 | 33.3 | 35.3 |
| MoCo (He et al., 2020) | 77.1 | 46.8 | 52.5 | 82.5 | 57.4 | 64.0 | 58.9 | 39.3 | 42.5 | 55.8 | 34.4 | 36.5 |
| BYOL (Grill et al., 2020b) | 77.1 | 47.0 | 49.9 | 81.4 | 55.3 | 61.1 | 57.8 | 37.9 | 40.9 | 54.3 | 33.2 | 35.0 |
| SimSiam (Chen & He, 2021) | 77.3 | 48.5 | 52.5 | 82.4 | 57.0 | 63.7 | 59.3 | 39.2 | 42.1 | 56.0 | 34.4 | 36.7 |
| SwAV (Caron et al., 2020) | 75.5 | 46.5 | 49.6 | 82.6 | 56.1 | 62.7 | 58.6 | 38.4 | 41.3 | 55.2 | 33.8 | 35.9 |
| MEC (Liu et al., 2022a) | 77.4 | 48.3 | 52.3 | 82.8 | 57.5 | 64.5 | 59.8 | 39.8 | 43.2 | 56.3 | 34.7 | 36.8 |
| VICRegL (Bardes et al., 2022) | 75.9 | 47.4 | 52.3 | 82.6 | 56.4 | 62.9 | 59.2 | 39.8 | 42.1 | 56.5 | 35.1 | 36.8 |
| SimCLR + Ours | 77.6 | 50.1 | 51.7 | 85.3 | 58.4 | 63.9 | 59.2 | 40.6 | 43.9 | 57.1 | 35.9 | 37.1 |
| MoCo + Ours | 79.4 | 50.2 | **54.9** | **86.1** | **60.2** | **66.1** | 61.4 | 42.1 | 44.9 | **59.2** | 36.9 | 38.8 |
| BYOL + Ours | 79.1 | 50.4 | 51.9 | 83.9 | 58.7 | 64.1 | 60.6 | 39.9 | 43.7 | 56.2 | 35.1 | 38.6 |
| SimSiam + Ours | **80.5** | **50.8** | 54.4 | 85.2 | 59.5 | 66.1 | 62.3 | 42.5 | 43.9 | 58.1 | 37.2 | 39.8 |
| SwAV + Ours | 77.9 | 49.3 | 51.8 | 84.9 | 58.1 | 65.8 | 62.1 | 40.2 | 43.9 | 56.9 | 37.3 | 37.9 |
| VICRegL + Ours | 77.9 | 50.4 | 53.9 | 85.2 | 58.8 | 65.3 | **63.1** | **42.2** | **45.3** | 59.1 | **37.8** | **39.9** |

wide range of batch sizes from 256 to 4096, and only drops for smaller values. Thus, we can obtain that although the proposed sampling strategy has a high requirement on a, the SSL method is less sensitive to a compared to the original batch size, which implies the effectiveness of our strategy.

**Influence of $\alpha$.** In Equation 4, $\alpha$ as a hyperparameter, controls the weight of the term that constrains the orthogonality of the column vectors in the matrix A. This constraint prevents the model from learning redundant or interdependent features, enhancing its generalization and stability. To evaluate its impact, we assess the performance of SimCLR+Ours and MoCo+Ours with varying $\alpha$ (ranging in $[0.001, 0.01, 0.1, 1, 10]$) on ImageNet-100, using the same configurations as in SSL. The results in Figure 5 show that performance peaks at $\alpha = 1$, which is also our setting.

# 6 RELATED WORK

**SSL** is an effective unsupervised representation learning paradigm, aimed at learning general representations suitable for various downstream tasks. From (Jaiswal et al., 2020; Kang et al., 2023), existing SSL models can be divided into two main types, i.e., D-SSL and G-SSL. The D-SSL methods, e.g., SimCLR (Chen et al., 2020), BYOL (Grill et al., 2020a), Barlow Twins (Zbontar et al., 2021), DINO (Caron et al., 2021), and Mocov3 (Chen et al., 2021b), are modeled based on the augmentation invariance principle. The G-SSL methods, e.g., MAE (He et al., 2022), VideoMAE (Tong et al., 2022), iBOT (Zhou et al.), SMA (Xie et al., 2024), are modeled based on the mask and reconstruction principle. In real-world scenarios, the data distribution can shift over time. Thus, improving the OOD generalization of SSL is crucial. Ni et al. (Ni et al., 2021) proposed to increase OOD generalization of SSL by meta-learning. MEC (Liu et al., 2022b) presents that a generalizable representation should be the one that admits the maximum entropy. AugSelf (Lee et al., 2021) encourages to preserve augmentation-aware information, which could be beneficial for feature transferability. KRR-ST (Lee et al., 2023) finds that distillation of SSL features using external knowledge

Table 4: Few-shot transfer learning accuracies ($\pm$ 95% confidence interval) on miniImageNet, Omniglot, and CIFAR-FS datasets with C4 as the backbone.

| Method | Omniglot | | | miniImageNet | | | CIFAR-FS | | |
|---|---|---|---|---|---|---|---|---|---|
| | (5,1) | (5,5) | (20,1) | (5,1) | (5,5) | (20,1) | (5,1) | (5,5) | (20,1) |
| SimCLR (Chen et al., 2020) | $90.83 \pm 0.21$ | $97.67 \pm 0.21$ | $81.67 \pm 0.23$ | $42.32 \pm 0.38$ | $51.10 \pm 0.37$ | $36.36 \pm 0.36$ | $49.44 \pm 0.30$ | $60.02 \pm 0.29$ | $39.29 \pm 0.30$ |
| MoCo (He et al., 2020) | $87.83 \pm 0.20$ | $95.52 \pm 0.19$ | $80.03 \pm 0.21$ | $40.56 \pm 0.34$ | $49.41 \pm 0.37$ | $36.52 \pm 0.38$ | $45.35 \pm 0.31$ | $58.11 \pm 0.32$ | $37.89 \pm 0.32$ |
| SwAV (Caron et al., 2020) | $91.28 \pm 0.19$ | $97.21 \pm 0.20$ | $82.02 \pm 0.20$ | $44.39 \pm 0.36$ | $54.91 \pm 0.36$ | $37.13 \pm 0.37$ | $49.39 \pm 0.29$ | $62.20 \pm 0.30$ | $40.19 \pm 0.32$ |
| SimCLR + Ours | $95.05 \pm 0.22$ | $\mathbf{98.96} \pm 0.16$ | $91.15 \pm 0.20$ | $47.14 \pm 0.21$ | $62.88 \pm 0.21$ | $39.97 \pm 0.16$ | $\mathbf{53.18} \pm 0.24$ | $67.91 \pm 0.14$ | $46.94 \pm 0.21$ |
| MoCo + Ours | $93.22 \pm 0.21$ | $97.93 \pm 0.19$ | $88.93 \pm 0.22$ | $46.93 \pm 0.21$ | $61.22 \pm 0.21$ | $41.12 \pm 0.24$ | $51.76 \pm 0.22$ | $66.42 \pm 0.21$ | $44.93 \pm 0.23$ |
| SwAV + Ours | $\mathbf{96.24} \pm 0.26$ | $98.76 \pm 0.22$ | $\mathbf{91.96} \pm 0.21$ | $\mathbf{49.15} \pm 0.21$ | $\mathbf{64.28} \pm 0.29$ | $\mathbf{42.22} \pm 0.21$ | $52.64 \pm 0.24$ | $\mathbf{70.18} \pm 0.21$ | $\mathbf{48.19} \pm 0.14$ |

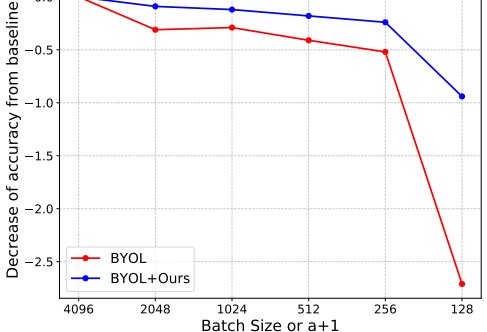 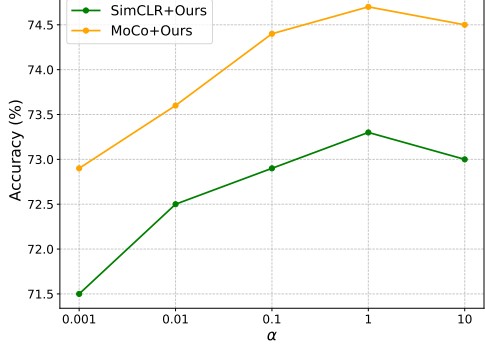

Figure 4: Influence of the hyperparameter $a$.      Figure 5: Influence of the hyperparameter $\alpha$.

can effectively improve OOD generalization. COLT (Bai et al., 2023) attempts to extend additional training samples from OOD datasets for improved SSL long-tailed learning. While various methods have been proposed with impressive performance, a remaining challenge is these approaches have to contend with trade-offs between inductive biases or approaches without theoretical guarantees. In this paper, we extend the understanding of SSL by analyzing its OOD generalization through the lens of causal inference and batch construction. Our proposed method addresses the limitations of existing approaches and offers a new direction for enhancing the OOD generalization of SSL.

**Causality Analysis in SSL** plays a crucial role by helping to identify and understand the underlying relationships between variables. Recent works Sontakke et al. (2021); Zuo et al. (2021); Qiang et al. (2022); Wang et al. (2024a) have focused on developing methods that leverage causal inference to extract more robust feature representations. For instance, Song et al. (2023) used causal invariance to obtain causal SSL representations and improve learning efficiency. Von Kügelgen et al. (2021) studied the identifiability of latent representations based on paired views of observations to study the effect of data augmentation performed in practice. However, most of them build causal analysis on in-distribution, but ignore the influence of spurious correlations under OOD generalization settings. In this paper, we explore the essential reasons for spurious correlations in SSL and propose a method that makes the relationships between variables free from the influence of spurious correlations.

## 7 CONCLUSION

In this paper, we focus on the OOD generalization of SSL models. First, we establish the connection between mini-batches formed during the SSL training phase and multi-class tasks. Next, we explain the rationale for OOD generalization of SSL from a multi-task learning perspective. We then analyze how existing SSL models, when learning mini-batch tasks, rely on spurious correlations to measure sample similarity, leading to suboptimal performance. This reliance affects the SSL model's approximation of the task distribution, resulting in reduced OOD generalization. We provide a causal analysis of this issue and theoretically examine the intrinsic reasons for incorporating spurious correlations during the learning process. Based on our causal analysis, we demonstrate that when mini-batches satisfy a specific distribution, e.g., PID, SSL models achieve optimal worst-case OOD performance. This insight guides us to propose a new mini-batch sampling strategy that ensures the resulting mini-batches satisfy the PID constraints. We provide a theoretical analysis of the effectiveness of this method and validate its efficacy through various downstream tasks.

## REPRODUCIBILITY STATEMENT

For the theoretical results, this work offers clear assumptions and complete proofs in the **Appendix**. The algorithm's source code is also submitted as supplementary materials. For the experimental datasets, detailed data processing steps and the experimental setup are provided in the **Appendix**.

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

APPENDIX

The **Appendix** provides supplementary material and additional details to support the main findings and methods proposed in this paper. It is organized into several sections:

- **Appendix** A contains the proofs of the presented theorems.

- **Appendix** B provides details for the experimental settings for each experiment.

- **Appendix** C showcases additional experiments that were omitted in the main text due to page limitations.

- **Appendix** D provides the related works for spurious correlation in SSL.

- **Appendix** E explains the differences and connections between task distribution and data distribution.

- **Appendix** F provides the intuitive explanation of several concepts, assumption, and theorems mentioned in the proposed methodology.

- **Appendix** G provides explanation of the identifiability of spurious variable.

## A  PROOFS

This section provides the complete proof of Proposition and Theorem in the main text.

### A.1  PROOF OF PROPOSITION 3.1

Before giving the detailed proofs of Proposition 3.1, we first provide the problem definition. Given multiple pairs of samples in an SSL task, let $x^{\text{label}}$ be the anchor of a specific pair, then the remaining samples involving two classes of being $x^{\text{label}}$ and not $x^{\text{label}}$. Let $x^{\text{label}}$ and $\bar{x}^{\text{label}}$ represent the label variables of being $x^{\text{label}}$ and not $x^{\text{label}}$, since these are binary classification tasks, $x^{\text{label}}$ and $\bar{x}^{\text{label}}$ belong to the set $\pm 1$. Note that any multi-classification task can be decomposed into binary tasks.

We assume that the labels are drawn from two different probabilities, with balanced sampling probabilities for label values, i.e., $P(x^{\text{label}} = 1) = P(x^{\text{label}} = -1) = 0.5$. Our conclusions also hold for imbalanced distributions. Next, we consider two $d$-dimensional factors $F_{x+}$ and $F_s$ representing the knowledge to tackle the two labels. Both are drawn from the Gaussian distribution:

$$F_{x+} \sim \mathcal{N}(x^{\text{label}} \cdot \mu_{\text{label}}, \sigma^2_{\text{label}} I)$$

$$F_s \sim \mathcal{N}(\bar{x}^{\text{label}} \cdot \mu_s, \sigma^2_s I)$$

where $\mu_{\text{label}}, \mu_s \in \mathbb{R}^{N_s}$ denote the mean vectors, while $\sigma^2_{\text{label}}$ and $\sigma^2_s$ denote the covariance vectors. We examine the spurious correlations in SSL. To simplify our analysis, we define $p_{sc}$ as the varying correlations that result from different spurious correlations across batches.

**Proposition 3.1** *Revisiting SSL from a pairwise perspective and assuming that the two samples in each pair satisfy Equation* (1), *we can obtain that the learned SSL model will use non-causal factor, i.e., the unobserved latent variable s, to measure the similarity or reconstruct in a pair.*

**Proofs:** Training a single model will result in the optimal model for the target incorporating non-causal features from the other sample pairs. To substantiate this, we derive the optimal SSL model as follows:

$$P(x^{\text{label}}|F_{x+}, F_s) = \frac{P(x^{\text{label}}, F_{x+}, F_s)}{P(F_{x+}, F_s)}$$

$$= \frac{P(x^{\text{label}}, F_{x+}, F_s)}{\sum_{x^{\text{label}} \in \{-1, 1\}} P(x^{\text{label}}, F_{x+}, F_s)}$$

where the probability $P(x^{\text{label}}, F_{x^+}, F_s)$ can be written as:

$$P(x^{\text{label}}, F_{x^+}, F_s) = P(x^{\text{label}}, F_{x^+}) \cdot P(F_s | x^{\text{label}}, F_{x^+})$$

$$= P(x^{\text{label}}, F_{x^+}) \cdot P(F_s | x^{\text{label}})$$

$$= P(x^{\text{label}}, F_{x^+}) \cdot \sum_{\bar{x}^{\text{label}} \in \{-1, 1\}} P(F_s, \bar{x}^{\text{label}} | x^{\text{label}})$$

$$= P(x^{\text{label}}) P(F_{x^+} | x^{\text{label}}) \cdot \sum_{\bar{x}^{\text{label}} \in \{-1, 1\}} P(F_s | \bar{x}^{\text{label}}) P(\bar{x}^{\text{label}} | x^{\text{label}})$$

Assuming that $F_{x^+}$ and $F_s$ are drawn from Gaussian distributions, and $P(Y_{i/j}, F_{x^+}, F_s) = \text{sigmoid}\left(\frac{\mu_{\text{label}}}{\sigma_{\text{label}}^2} F_{x^+} + \frac{\mu_s}{\sigma_s^2} F_s\right)$, where $\frac{\mu_{\text{label}}}{\sigma_{\text{label}}^2}$ and $\frac{\mu_s}{\sigma_s^2}$ are the regression vectors for the optimal Bayesian classifier, we have:

$$P(x^{\text{label}}, F_{x^+}, F_s) = P(x^{\text{label}}, F_{x^+}) \cdot P(F_s | x^{\text{label}}, F_{x^+})$$

$$= P(x^{\text{label}}) P(F_{x^+} | x^{\text{label}}) \cdot \sum_{\bar{x}^{\text{label}} \in \{-1, 1\}} P(F_s | \bar{x}^{\text{label}}) P(\bar{x}^{\text{label}} | x^{\text{label}})$$

$$\propto e^{x^{\text{label}} \cdot \frac{\mu_{\text{label}}}{\sigma_{\text{label}}^2} F_{x^+}} \left( p_{sc} e^{x^{\text{label}} \cdot \frac{\mu_s}{\sigma_s^2} F_s} + (1 - p_{sc}) e^{-x^{\text{label}} \cdot \frac{\mu_s}{\sigma_s^2} F_s} \right)$$

$$= p_{sc} e^{x^{\text{label}} \cdot \left( \frac{\mu_{\text{label}}}{\sigma_{\text{label}}^2} F_{x^+} + \frac{\mu_s}{\sigma_s^2} F_s \right)} + (1 - p_{sc}) e^{x^{\text{label}} \cdot \left( \frac{\mu_{\text{label}}}{\sigma_{\text{label}}^2} F_{x^+} - \frac{\mu_s}{\sigma_s^2} F_s \right)}$$

Let:

$$\beta^+ = \frac{\mu_{\text{label}}}{\sigma_{\text{label}}^2} F_{x^+} + \frac{\mu_s}{\sigma_s^2} F_s$$

$$\beta^- = \frac{\mu_{\text{label}}}{\sigma_{\text{label}}^2} F_{x^+} - \frac{\mu_s}{\sigma_s^2} F_s$$

Substituting $\beta^+$ and $\beta^-$ back into the original equation, we have:

$$P(x^{\text{label}} | F_{x^+}, F_s) = \frac{1}{1 + \frac{p_{sc} e^{x^{\text{label}} \cdot \beta^+} + (1 - p_{sc}) e^{x^{\text{label}} \cdot \beta^-}}{p_{sc} e^{-x^{\text{label}} \cdot \beta^+} + (1 - p_{sc}) e^{-x^{\text{label}} \cdot \beta^-}}}$$

When the samples are easy to distinguish, e.g., the similarity of the augmented sample from different pairs is not 1:

$$P(x^{\text{label}} | F_{x^+}, F_s) = \frac{1}{1 + e^{x^{\text{label}} \cdot (\beta^+ + \beta^-)}}$$

Combining with the expressions for $\beta^+$ and $\beta^-$, we get:

$$P(x^{\text{label}} | F_{x^+}, F_s) = \frac{1}{1 + e^{2x^{\text{label}} \cdot \left( \frac{\mu_{\text{label}}}{\sigma_{\text{label}}^2} F_{x^+} \right)}}$$

In this case, the optimal SSL model only utilizes its own factor $F_{x^+}$ and assigns zero weight to the non-causal factor $F_s$ from task $\tau_j$. Thus, if it is difficult to distinguish between the different pairs, the optimal model has non-zero weights for non-causal factors for each task.

When the samples are difficult to distinguish, e.g., in the most extreme case, the similarity of the augmented sample from different pairs is equal to 1, we have:

$$P(x^{\text{label}} | F_{x^+}, F_s) = \frac{1}{1 + e^{2x^{\text{label}} \cdot \beta^+}}$$

Combining with the expressions for $\beta^+$ and $\beta^-$, we get:

$$P(x^{\text{label}} | F_{x^+}, F_s) = \frac{1}{1 + e^{2x^{\text{label}} \cdot \left( \frac{\mu_{\text{label}}}{\sigma_{\text{label}}^2} F_{x^+} + \frac{\mu_s}{\sigma_s^2} F_s \right)}}$$

In this case, the optimal classifier incorporates both factors $F_{x^+}$ and $F_s$. Thus, if $p_{sc} \neq 0.5$, the optimal classifier assigns non-zero weights to non-causal factors for each task.

## A.2 PROOF OF THEOREM 3.4

**Theorem 3.4** *From a Bayesian perspective, the alignment part of the SSL learning objective, e.g., constrain samples under the same pair to be similar in the feature space, can be expressed as* $\max p_f(x^{\text{label}}|x^+)$. *Given* $f$, *the risk on a batch with* $e \in \mathcal{D}$ *as the distributional constraint can be presented as:* $\mathcal{L}^e(f) = \mathbb{E}_{p^e(x^+,x^{\text{label}})} - \log p_f(x^{\text{label}}|x^+)$, *where* $p^e(x^+, x^{\text{label}})$ *denotes the joint distribution. Under **Assumption 3.3**, when* $f^* = \arg\max \mathcal{L}^{\text{PID}}(f)$, *we have* $f^*$ *is the minimax optimal across all elements in* $\mathcal{D}$, *e.g.,* $f^* = \arg_f \min\max_{e \in \mathcal{D}} \mathcal{L}^e(p_f(x^{\text{label}}|x^+))$.

**Proofs:** Here, we provide proof of the minimax optimality of the SSL model trained on PID. The SSL model trained on PID $p^{\text{PI}}(x^+, x^{\text{label}})$ has $p_f(x^{\text{label}}|x^+) = p^{\text{PI}}(x^{\text{label}}|x^+)$. Now, consider the expected cross-entropy loss of this classifier on an unseen test distribution $p^e$:

$$\mathcal{L}^e(p^{\text{PI}}(x^{\text{label}}|x^+)) = -\mathbb{E}_{p^e(x^+,x^{\text{label}})} \log p^{\text{PI}}(x^{\text{label}}|x^+)$$

$$= -\mathbb{E}_{p^e(x^+,x^{\text{label}})} \log p^{\text{PI}}(x^{\text{label}}) + \mathbb{E}_{p^e(x^+,x^{\text{label}})} \log \frac{p^{\text{PI}}(x^{\text{label}})}{p^{\text{PI}}(x^{\text{label}}|x^+)}$$

$$= \mathcal{L}^e(p^{\text{PI}}(x^{\text{label}})) + \mathbb{E}_{p^e(X,x^{\text{label}},s)} \left[ \log \frac{p^{\text{PI}}(x^{\text{label}})}{p^{\text{PI}}(x^{\text{label}}|x^+)} \right]$$

$$= \mathcal{L}^e(p^{\text{PI}}(x^{\text{label}})) + \mathbb{E}_{p^e(x^{\text{label}},s)} \left[ \mathbb{E}_{p^{\text{PI}}(X|x^{\text{label}},s)} \left[ \log \frac{p^{\text{PI}}(x^{\text{label}})}{p^{\text{PI}}(x^{\text{label}}|x^+)} \right] \right]$$

Consider that $x^{\text{label}} \perp\!\!\!\perp_{\text{PI}} s$ and $x^{\text{label}} \perp\!\!\!\perp_{\text{PI}} s|x^+$, we get:

$$\mathcal{L}^e(p^{\text{PI}}(x^{\text{label}}|x^+)) = \mathcal{L}^e(p^{\text{PI}}(x^{\text{label}})) + \mathbb{E}_{p^e(x^{\text{label}},s)} \left[ \mathbb{E}_{p^{\text{PI}}(x^+|x^{\text{label}},s)} \left[ \log \frac{p^{\text{PI}}(x^{\text{label}}|s)}{p^{\text{PI}}(x^{\text{label}}|x^+,s)} \right] \right]$$

$$= \mathcal{L}^e(p^{\text{PI}}(x^{\text{label}})) + \mathbb{E}_{p^e(x^{\text{label}},s)} \left[ \mathbb{E}_{p^{\text{PI}}(x^+|x^{\text{label}},s)} \left[ \log \frac{p^{\text{PI}}(x^+|s)}{p^{\text{PI}}(x^+|x^{\text{label}},s)} \right] \right]$$

$$= \mathcal{L}^e(p^{\text{PI}}(x^{\text{label}})) - \mathbb{E}_{p^e(x^{\text{label}},s)} KL[p^{\text{PI}}(x^+|x^{\text{label}},s)||p^{\text{PI}}(x^+|s)].$$

Thus we have the cross entropy loss of $p^{\text{PI}}(x^+, x^{\text{label}})$ in any environment $e$ is smaller than that of $p^{\text{PI}}(x^{\text{label}}) = \frac{1}{m}$ (random guess):

$$\mathcal{L}^e(p^{\text{PI}}(x^{\text{label}}|x^+)) - \mathcal{L}^e(p^{\text{PI}}(x^{\text{label}})) \leq -\mathbb{E}_{p^e(x^{\text{label}},s)} KL[p^{\text{PI}}(x^+|x^{\text{label}},s)||p^{\text{PI}}(x^+|s)] \leq 0,$$

which means:

$$\max_{e' \in \mathcal{E}} \left[ L^{e'}(p^{\text{PI}}(x^{\text{label}}|x^+)) - L^{e'}(p^{\text{PI}}(x^{\text{label}})) \right] \leq 0.$$

where the performance of $p^{\text{PI}}(x^+, x^{\text{label}})$ is at least as good as a random guess in any environment. Since we assume the environment diversity, that is for any $p^e$ with $x^{\text{label}} \perp\!\!\!\perp_e s$, there exists an environment $e'$ such that $p^e(x^{\text{label}}|x^+)$ performs worse than a random guess. So we have:

$$\max_{e' \in \mathcal{E}} \left[ \mathcal{L}^{e'}(p^{\text{PI}}(x^{\text{label}}|x^+)) - \mathcal{L}^{e'}(p^{\text{PI}}(x^{\text{label}})) \right] \leq 0 < \max_{e' \in \mathcal{E}} \left[ \mathcal{L}^{e'}(p^e(x^{\text{label}}|x^+)) - \mathcal{L}^{e'}(p^{\text{PI}}(x^{\text{label}})) \right].$$

Now we want to prove that $\forall e \in \mathcal{E}$, $x^{\text{label}} \perp\!\!\!\perp_e s$, $x^{\text{label}} \perp\!\!\!\perp_e s|x^+$, $p^e(x^{\text{label}}) = \frac{1}{m}$ $\implies$ $p^e(x^{\text{label}}|x^+) = p^{\text{PI}}(x^{\text{label}}|x^+)$. For any $s \in \mathcal{S}$, we have:

$$p^e(x^{\text{label}}|x^+) = p^e(x^{\text{label}}|x^+, s)$$

$$= p^e(x^{\text{label}}) \frac{p^e(x^+|x^{\text{label}}, s)}{\mathbb{E}_{p^e(x^{\text{label}}|s)}[p^e(x^+|s, x^{\text{label}})]}$$

$$= p^{\text{PI}}(x^{\text{label}}) \frac{p^{\text{PI}}(x^+|x^{\text{label}}, s)}{\mathbb{E}_{p^{\text{PI}}(x^{\text{label}})}[p^{\text{PI}}(x^+|s, x^{\text{label}})]}$$

$$= p^{\text{PI}}(x^{\text{label}}|x^+, s) = p^{\text{PI}}(x^{\text{label}}|x^+).$$

Thus we have the following minimax optimality:

$$p^{\text{PI}}(x^{\text{label}}|x^+) = \arg\min_{p_f \in \mathcal{F}} \max_{e \in \mathcal{E}} \mathcal{L}^e(p_\psi(x^{\text{label}}|x^+)).$$

Thus, we have $f^*$ is the minimax optimal across all elements in $\mathcal{D}$, e.g., $f^* = \arg_f \min\max_{e \in \mathcal{D}} \mathcal{L}^e(p_f(x^{\text{label}}|x^+))$.

## A.3 Proof of Theorem 4.3

**Theorem 4.3** *Suppose that $p_\theta^e(x^+, s|x^{\text{label}}) = p_f(x^+|s, x^{\text{label}})p_{g,A}(s|x^{\text{label}})$ and the generation process of $X^+$ can be represented by the SCM depicted in Figure 1, a sufficient condition for $\theta = (f, g, A)$ to be $\sim_A$-identifiable is given as: 1) Suppose that $p_\epsilon(x^+ - f(x^{\text{label}}, s)) = p_f(x^+|x^{\text{label}}, s)$, $\phi_\varepsilon$ is the characteristic function of $p_\epsilon(x^+ - f(x^{\text{label}}, s))$, and the set $\{x^+|\phi_\varepsilon(x^+) = 0\}$ has measure zero; 2) The sufficient statistics $T$ are differentiable almost everywhere, and $[T_{ij}]_{1 \le j \le k}$ are linearly independent on any subset of $X^+$ with measure greater than zero; 3) There exist $nk + 1$ distinct pairs $(x_0^{\text{label}}, e_0), \cdots, (x_n^{\text{label}}k, e_{nk})$ such that the $nk \times nk$ matrix $L = (\lambda^{e_1}(x_1^{\text{label}}) - \lambda^{e_0}(x_0^{\text{label}}), \cdots, \lambda^{e_{nk}}(x_{nk}^{\text{label}}) - \lambda^{e_0}(x_0^{\text{label}}))$ is invertible.*

**Proofs:** We now establish Theorem 4.3, demonstrating the identifiability of the essential parameters that capture spuriously correlated covariate features in the VAE. The proof consists of three steps: (i) We use both $e$ and $x^{\text{label}}$ as auxiliary variables; (ii) We include $x^{\text{label}}$ in the causal mechanism of generating $x^+$ by $x = f(x^{\text{label}}, s) + \epsilon = f_x^{\text{label}}(x) + \epsilon$.

First, we transform the equality of the marginal distributions over the observed data into the equality of a noise-free distribution. Suppose we have two sets of parameters, $\theta = (f, g, A)$ and $\theta' = (f', g', A')$, such that $p_\theta(x^+|x^{\text{label}}, e) = p_{\theta'}(x^+|x^{\text{label}}, e)$ for all $e \in \mathcal{E}_{\text{train}}$. Then:

$$\int_{\mathcal{Z}} p_{g,A}(Z|x^{\text{label}}, e)p_f(x^+|Z, x^{\text{label}})dZ = \int_{\mathcal{Z}} P_{g',A'}(Z|x^{\text{label}}, e)p'_f(x^+|Z, x^{\text{label}})dZ$$

$$\int_{\mathcal{Z}} p_{g,A}(Z|x^{\text{label}}, e)p_\epsilon(x^+ - f_x^{\text{label}}(Z))dZ = \int_{\mathcal{Z}} p_{g',A'}(Z|x^{\text{label}}, e)p_\epsilon(x^+ - f_x^{'\text{label}}(Z))dZ$$

$$(6)$$

Then, we denote the volume of a matrix A as $\text{vol}A := \sqrt{\det(A^\top A)}$, $J$ as the Jacobian, and change the variable on the left-hand side to $x^+ = f_x^{\text{label}}(Z)$ and on the right-hand side to $\bar{x}^+ = \bar{f}_x^{\text{label}}(Z)$. Since f is injective, we have $f^{-1}(\bar{x}^+) = (x^{\text{label}}, Z)$. Here, we specifically use $f^{-1}(\bar{x}^+)$ to denote the recovery of $Z$, i.e., $f^{-1}(\bar{x}^+) = Z$. Then, we get:

$$\int_{\mathbb{R}^d} \tilde{p}_{g,A,f,x^{\text{label}},e}(\bar{x}^+)p_\epsilon(x^+ - \bar{x}^+)d\bar{x}^+ = \int_{\mathbb{R}^d} \tilde{p}_{g',A',f',x^{\text{label}},e}(\bar{x}^+)p_\epsilon(x^+ - \bar{x}^+)d\bar{x}^+ \tag{7}$$

$$(8)$$

Next, we introduce

$$\tilde{p}_{g,A,f,x^{\text{label}},e}(x^+) = p_{g,A}(f_x^{\text{label}^{-1}}(x^+)|x^{\text{label}}, e)\text{vol}J_{f_x^{\text{label}-1}}(x^+)\mathbb{1}_{\S^+}(x^+),$$

on the left-hand side, and similarly on the right-hand side:

$$(\tilde{p}_{g,A,f,x^{\text{label}},e} * p_\epsilon)(x^+) = (\tilde{p}_{g',A',f',x^{\text{label}},e} * P_\mathcal{E})(x^+) \tag{9}$$

$$(10)$$

Then, we use $*$ for the convolution operator, and use $F[\cdot]$ to designate the Fourier transform. The characteristic function of $\epsilon$ is then $\phi_\epsilon = F[p_\epsilon]$. Exploit the properties of the Fourier transform to transform the convolution into a multiplication. This means that in the Fourier domain, we have $F[(\tilde{p}_{g,A,f,x^{\text{label}},e} * p_\epsilon)(x^+)] = F[\tilde{p}_{g,A,f,x^{\text{label}},e}](\omega) \cdot F[p_\epsilon](\omega)$ Meanwhile, we dropped $\phi_\epsilon(\omega)$ from both sides as it is non-zero almost everywhere (by assumption of the Theorem).

$$\tilde{p}_{g,A,f,x^{\text{label}},e}(x^+) = \tilde{p}_{g',A',f',x^{\text{label}},e}(x^+). \tag{11}$$

For the second step, in this step, we remove all terms that are either a function of $x^+$ or $x^{\text{label}}$ or $e$. By taking logarithm on both sides of Equation 11 and replacing $p_{g,A}$ by its expression, we get:

$$\log \text{vol}J_{f^{-1}}(x^+) + \sum_{i=1}^n (\log Q_i(f_i^{-1}(x^+)) - \log W_i^e(x^{\text{label}}) + \sum_{j=1}^k T_{i,j}(f_i^{-1}(x^+))\lambda_{i,j}^e(x^{\text{label}}))$$

$$= \log \text{vol}J_{f'^{-1}}(x^+) + \sum_{i=1}^n (\log Q'_i(f_i'^{-1}(x^+)) - \log W_i'^e(x^{\text{label}}) + \sum_{j=1}^k T'_{i,j}(f_i'^{-1}(x^+))\lambda_{i,j}'^e(x^{\text{label}})).$$

Let $(e_0, x_0^{\text{label}}), (e_1, x_1^{\text{label}}), ..., (e_{nk}, x_{nk}^{\text{label}})$ be the points provided by assumption (3) of the Theorem. We evaluate the above equations at these points to obtain $k + 1$ equations, and subtract the first equation from the remaining $k$ equations to obtain:

$$\langle T(f^{-1}(x^+)), \lambda^{e_l}(x_l^{\text{label}}) - \lambda^{e_0}(x_0^{\text{label}})\rangle + \sum_{i=1}^n \log \frac{W_i^{e_0}(x_0^{\text{label}})}{W_i^{e_l}(x_l^{\text{label}})}$$

$$= \langle T'(f'^{-1}(x^+)), \lambda'^{e_l}(x_l^{\text{label}}) - \lambda'^{e_0}(x_0^{\text{label}})\rangle + \sum_{i=1}^n \log \frac{W_i'^{e_0}(x_0^{\text{label}})}{W_i'^{e_l}(x_l^{\text{label}})}. \tag{12}$$

Let $\mathcal{L}$ be the matrix defined in assumption (3) and $\mathcal{L}'$ similarly defined for $\lambda'$ ($\mathcal{L}'$ is not necessarily invertible). Define $b_l = \sum_{i=1}^n \log \frac{W_i'^{e_0}(x_0^{\text{label}})W_i^{e_l}(x_l^{\text{label}})}{W_i^{e_0}(x_0^{\text{label}})W_i'^{e_l}(x_l^{\text{label}})}$ and $b = [b_l]_{l=1}^{nk}$.

Then Equation 12 can be rewritten in the matrix form:

$$\mathcal{L}^T T(f^{-1}(x^+)) = \mathcal{L}'^T T'(f'^{-1}(x^+)) + b. \tag{13}$$

We multiply both sides of Equation 13 by $\mathcal{L}^{-T}$ to get:

$$T(f^{-1}(x^+)) = A T'(f'^{-1}(x^+)) + c. \tag{14}$$

Where $A = \mathcal{L}^{-T}\mathcal{L}'$ and $c = \mathcal{L}^{-T}b$. To complete the proof, we must demonstrate that A is invertible. By the definition of $T$, its Jacobian exists and is an $nk \times n$ matrix with rank $n$. Consequently, the Jacobian of $T' \circ f'^{-1}$ also exists and has rank $n$, which implies that $A$ is of rank $n$ as well. We mainly consider two cases:

If $k = 1$, then A is invertible since $A \in \mathbb{R}^{n \times n}$.

If $k > 1$, define $\bar{\mathbf{x}} = f^{-1}(\mathbf{x})$ and $T_i(\bar{x}_i) = (T_{i,1}(\bar{x}_i), \ldots, T_{i,k}(\bar{x}_i))$.

Suppose for any choice of $\bar{x}_i^1, \bar{x}_i^2, \ldots, \bar{x}_i^k$, the family $\left(\frac{dT_i(\bar{x}_i^1)}{d\bar{x}_i^1}, \ldots, \frac{dT_i(\bar{x}_i^k)}{d\bar{x}_i^k}\right)$ is never linearly independent. This implies that $T_i(\mathbb{R})$ lies within a subspace of $\mathbb{R}^k$ with a dimension of at most $k - 1$. Let $h$ be a non-zero vector orthogonal to $T_i(\mathbb{R})$. Then for all $x \in \mathbb{R}$, we have $\left\langle \frac{dT_i(x)}{dx}, h \right\rangle = 0$. By integrating, we find that $\langle T_i(x), h \rangle = \text{const}$.

Since this holds for all $x \in \mathbb{R}$ and $h \neq 0$, we conclude that the distribution is not strongly exponential. Thus, by contradiction, there must exist $k$ points $\bar{x}_i^1, \bar{x}_i^2, \ldots, \bar{x}_i^k$ such that $\left(\frac{dT_i(\bar{x}_i^1)}{d\bar{x}_i^1}, \ldots, \frac{dT_i(\bar{x}_i^k)}{d\bar{x}_i^k}\right)$ are linearly independent.

Next, collect these points into $k$ vectors $(\bar{x}^1, \ldots, \bar{x}^k)$ and concatenate the $k$ Jacobians $J_T(\bar{x}^l)$ evaluated at each of those vectors horizontally into the matrix $Q = (J_T(\bar{x}^1), \ldots, J_T(\bar{x}^k))$. Similarly, define $Q'$ as the concatenation of the Jacobians of $T'(f'^{-1} \circ f(\bar{x}))$ evaluated at those points. Then the matrix $Q$ is invertible. By differentiating Equation 14 for each $x^l$, we get $Q = AQ'$ The invertibility of $Q$ implies the invertibility of A and $Q'$. This completes the proof.

## A.4    Proof of Theorem 4.7

**Theorem 4.7** *If $d(ba(s_j), ba(s_i)) = 0$ in **Algorithm** 1, the obtained mini-batch is regarded as sampling from a PID, e.g., $\hat{p}(x^{\text{label}}|s) = p^{\text{PI}}(x^{\text{label}})$.*

**Proofs:** In **Algorithm** 1, by uniformly sampling $a$ different labels, we mean sampling $x_{\text{alt}}^{\text{label}} = \{x_1^{\text{label}}, x_2^{\text{label}}, ..., x_a^{\text{label}}\}$ using the following procedure:

$$x_1^{\text{label}} \sim U\{1, 2, ..., mu\} \setminus \{x_e^{\text{label}}\}$$
$$x_2^{\text{label}} \sim U\{1, 2, ..., mu\} \setminus \{x_e^{\text{label}}, x_1^{\text{label}}\}$$
$$\vdots$$
$$x_a^{\text{label}} \sim U\{1, 2, ..., mu\} \setminus \{x_e^{\text{label}}, x_1^{\text{label}}, x_2^{\text{label}}, ..., x_{a-1}^{\text{label}}\},$$

where $U$ denotes the uniform distribution.

Suppose $\mathcal{D}_{\text{balanced}} \sim \hat{p}^B(x^+, x^{\text{label}})$, and the data distribution $\mathcal{D}^e \sim p(x^+, x^{\text{label}})$. Assume we have an exact match every time we match a balancing score. Then for all $e \in \mathcal{E}_{\text{train}}$, we have:

$$\hat{p}^B(x^{\text{label}}|ba^e(s)) = p(x^{\text{label}}|ba^e(s)). \tag{15}$$

By the definition of a balancing score, $p(x^{\text{label}}|s) = p(x^{\text{label}}|ba^e(s))$ and $\hat{p}^B(x^{\text{label}}|s) = \hat{p}^B(x^{\text{label}}|ba^e(s))$, then we have:

$$\hat{p}^B(x^{\text{label}}|s) = p(x^{\text{label}}|s).$$

Thus, we have $\hat{p}^B(x^{\text{label}}|s) = U\{1, 2, ..., mu\}$, which means $\hat{p}^B(x^+, x^{\text{label}}, s) = p^B(x^+, x^{\text{label}}, s)$. This implies that $\mathcal{D}_{\text{balanced}}$ can be regarded as sampled from a PID.

## B    EXPERIMENTAL SETTINGS

In this section, we provide the details of the settings and datasets for each experiment.

**Unsupervised Learning** Following the widely adopted protocol Chen et al. (2020); Wang et al. (2024b), we freeze the feature extractor and train a supervised linear classifier on top of it. The Adam optimizer is used, with Momentum set to 0.8 and weight decay set to $10^{-4}$. The linear classifier is trained for 500 epochs, with a batch size of 128. The learning rate starts at $5 \times 10^{-2}$ and decays to $5 \times 10^{-6}$. For this experiment, we utilize several benchmark datasets to evaluate the model's performance. CIFAR-10 and CIFAR-100 are small-scale image classification datasets consisting of 60,000 32×32 color images in 10 and 100 classes, respectively. STL-10 is another small-scale dataset that contains 100,000 unlabeled images and 5,000 labeled examples from 10 classes, with a higher image resolution (96×96). Tiny ImageNet contains 100,000 64×64 images across 200 classes and serves as a more challenging small-scale benchmark. For these datasets, we use ResNet-18 as the feature extractor. For larger datasets, we employ ImageNet-100 (a subset of ImageNet with 100 classes) and the full ImageNet dataset, which consists of over 1.2 million images in 1,000 classes, using ResNet-50 as the feature extractor.

**Semi-Supervised Learning** In accordance with the standard protocol Zbontar et al. (2021), we create two balanced subsets by sampling 1% and 10% of the training dataset. Specifically, we use the ImageNet dataset, a large-scale benchmark for visual recognition tasks, comprising 1.2 million images in 1,000 categories. The subsets contain 1% and 10% of the labeled training data, which are used for fine-tuning the model. The models are fine-tuned for 50 epochs, with learning rates set to 0.05 and 1.0 for the classifier and 0.0001 and 0.01 for the backbone on the 1% and 10% subsets, respectively.

**Transfer Learning** We conduct three transfer learning experiments, including object detection and instance segmentation, transfer to other domains, and video-based tasks. For object detection, we evaluate the model on two benchmark datasets: Pascal VOC and COCO. Pascal VOC is widely used for object detection tasks, containing around 20,000 images across 20 categories. We train a Faster R-CNN Ren et al. (2015) model on the combined VOC 2007 and 2012 datasets (VOC 07+12), which contains around 16,000 images, and adjust the learning rate at 18K and 22K iterations. We also conduct experiments on a smaller version of Pascal VOC, the VOC 07 set (5K images), with a reduced number of iterations. For instance segmentation, we use the COCO 2017 dataset, which contains over 118,000 images and covers 80 object categories. We train a Mask R-CNN He et al.

(2017) with the standard 1× schedule and C4-backbone Wu et al. (2019), reporting results on the validation split.

**Few-shot Learning** The protocol outlined in Wang et al. (2024b; 2023b) is followed for few-shot learning, where we evaluate the proposed method on three standard few-shot learning benchmarks: miniImageNet, Omniglot, and CIFAR-FS. miniImageNet is a widely used few-shot learning benchmark derived from the ImageNet dataset, consisting of 60,000 84×84 images across 100 classes. Omniglot is a dataset designed for character recognition, containing 1,623 different characters from 50 different alphabets, making it suitable for testing few-shot learning algorithms. CIFAR-FS is a few-shot version of the CIFAR-100 dataset, specifically adapted for few-shot learning tasks, containing 100 classes with 600 images per class. For each task, $N$ samples without class-level overlap are randomly selected, and $K$-times data augmentation is applied to create an $N$-way $K$-shot task. The model is optimized using stochastic gradient descent (SGD) with momentum and weight decay values set to 0.9 and $10^{-4}$, respectively. The trained model's performance is then evaluated on unseen samples drawn from new classes, testing its ability to generalize in few-shot scenarios.

# C ADDITIONAL EXPERIMENTS

## C.1 EVALUATION ON GENERATIVE SSL

To examine the model's impact on generating SSL, we conducted a series of experiments using the ImageNet-1K dataset (Deng et al., 2009). We started with self-supervised pre-training on the ImageNet-1K (IN1K) training set. Next, we evaluated the representations through supervised training using two methods: (i) end-to-end fine-tuning and (ii) linear probing. We reported the top-1 validation accuracy for a single 224×224 crop. For these experiments, we employed ViT-Large (ViT-L/16) (Dosovitskiy et al., 2020) as the backbone. ViT-Large is significantly larger (an order of magnitude bigger) than ResNet-50 (He et al., 2016) and has a tendency to overfit. The following section provides a comparison of the models.

Table 5: Comparison between models.

| Method | scratch, original | scratch, our impl. | baseline MAE | MAE + Ours |
|---|---|---|---|---|
| Top 1 | 76.5 | 82.5 | 84.9 | 86.4 |

Table 6: Comparisons with previous results on ImageNet-1K using the ImageNet-1K training set for pre-training, except for the tokenizer in BEiT, which was pre-trained on 250M DALLE data (Ramesh et al., 2021).

| Method | pre-train data | ViT-B | ViT-L | ViT-H | ViT-H$_{448}$ |
|---|---|---|---|---|---|
| DINO | IN1K | 82.8 | - | - | - |
| MoCo | IN1K | 83.2 | 84.1 | - | - |
| BEiT | IN1K+DALLE | 83.2 | 85.2 | - | - |
| MAE | IN1K | 83.6 | 85.9 | 86.9 | 87.8 |
| MAE+Ours | IN1K | 85.9 | 87.4 | 88.6 | 89.3 |

**Comparisons with self-supervised methods.** In **Table** 6 we compare the fine-tuning results of self-supervised ViT models. Our method has shown steady improvement from bigger models. We obtain 88.6% accuracy using ViT-H (224 size). The previous best accuracy, among all methods, using only IN1K data, is 87.1% (512 size) (Yuan et al., 2022), based on advanced networks. We improve over the state-of-the-art by a nontrivial margin in the highly competitive benchmark of IN1K (no external data). Our result is based on vanilla ViT, and we expect advanced networks will perform better.

**Object detection and segmentation.** We fine-tune Mask R-CNN (He et al., 2017) end-to-end on COCO (Lin et al., 2014). The ViT backbone is adapted for use with FPN (Lin et al., 2017). We apply this approach to all entries in Table 3. We report box AP for object detection and mask AP

Table 7: COCO object detection and segmentation using a ViT Mask R-CNN baseline.

| Method | pre-train data | AP$^{box}$ | | AP$^{mask}$ | |
|---|---|---|---|---|---|
| | | ViT-B | ViT-L | ViT-B | ViT-L |
| supervised | IN1K w/ labels | 47.9 | 49.3 | 42.9 | 43.9 |
| MoCo v3 | IN1K | 47.9 | 49.3 | 42.7 | 44.0 |
| BEiT | IN1K+DALLE | 49.8 | 53.3 | 44.4 | 47.1 |
| MAE | IN1K | 50.3 | 53.3 | 44.9 | 47.2 |
| MAE + Ours | IN1K | 52.5 | 55.9 | 46.4 | 49.7 |

Table 8: Performance on for text recognition.

| Methods | IIIT5K | IC03 |
|---|---|---|
| SimCLR Chen et al. (2020) | 1.7 | 3.8 |
| SeqCLR Aberdam et al. (2021) | 35.7 | 43.6 |
| SimCLR + Ours | 18.7 | 19.0 |
| SeqCLR + Ours | 38.5 | 47.4 |

for instance segmentation. Compared to supervised pre-training, our MAE performs better under all configurations (**Table** 7).

## C.2 EVALUATION ON MORE MODALITIES

The proposed method can be applied in various fields and domains, e.g., instance segmentation, video tracking, sample generation, etc., as mentioned before. Here, we provide the experiments of the proposed method on text modality-based datasets, i.e., IC03 and IIIT5K Yasmeen et al. (2020), which we have conducted before. We follow the same experimental settings as mentioned in Aberdam et al. (2021). The results shown in **Table** 8 demonstrate that the proposed method achieves stable effectiveness and robustness in various modalities combined with the above experiments.

Table 9: Performance comparison on PACS dataset.

| Method | Photo | Sketch | Cartoon | Painting (Unseen) | Average |
|---|---|---|---|---|---|
| SimCLR | 86.4 | 85.1 | 87.2 | 74.3 | 80.7 |
| SimCLR+Ours | 88.0 | 87.4 | 90.1 | 79.2 | 85.0 |
| BYOL | 83.9 | 84.6 | 82.7 | 64.5 | 74.2 |
| BYOL+Ours | 84.2 | 86.9 | 85.0 | 70.8 | 78.9 |

## C.3 EVALUATION ON OOD TASKS

In addition to validating the proposed method on standard and few-shot transfer learning scenarios, we also specifically test it on benchmark datasets targeting the out-of-distribution (OOD) problem, including PACS, OfficeHome, and ColoredMNIST. Specifically, we evaluate the performance of SSL baselines before and after introducing the proposed PID on these three datasets. For PACS, we follow the experimental setup in **Section 5.2** to evaluate the most commonly

Table 11: Performance comparison on ColoredMNIST dataset.

| Method | Accuracy(%) |
|---|---|
| SimCLR | 85.2 |
| SimCLR + Ours | 88.6 |

used SSL baselines, SimCLR and BYOL, on three domains—Photo, Sketch, and Cartoon—training on these domains and testing on all four domains, including Photo, Art, Cartoon, and Sketch, as well as the average performance. The results are shown in **Table 9**. For OfficeHome, we randomly select one domain as the source domain for training and another as the target domain. The labels for

Table 10: Performance comparison on OfficeHome dataset

| Method | A$\to$C | A$\to$P | A$\to$R | C$\to$A | C$\to$P | C$\to$R | P$\to$A | P$\to$C | P$\to$R | R$\to$A | R$\to$C | R$\to$P |
|---|---|---|---|---|---|---|---|---|---|---|---|---|
| SimCLR | 58.2 | 63.5 | 69.8 | 78.9 | 69.7 | 66.8 | 63.4 | 52.3 | 58.4 | 56.1 | 72.9 | 71.0 |
| SimCLR+Ours | 61.1 | 65.2 | 71.9 | 81.1 | 72.0 | 68.2 | 67.5 | 59.1 | 59.9 | 61.2 | 74.8 | 73.5 |

the source domain are predefined, whereas the labels for all target domains are unknown. We then evaluate the performance change of SimCLR before and after introducing PID, with results shown in **Table 10**. Finally, for ColoredMNIST, we follow the experimental setup in Gat et al. (2020), assessing the model's performance on new classes after training on base classes, with results shown in **Table 11**. The results demonstrate that PID consistently improves performance, confirming its effectiveness on OOD tasks.

## D  RELATED WORKS FOR SPURIOUS CORRELATION

In the recent work on SSL, there has been growing interest in understanding its vulnerability to spurious correlations Hamidieh et al. (2024); Wang et al. (2022; 2023a). These correlations arise when models learn associations from data that do not truly reflect the underlying causal structure, but instead are coincidental or context-specific patterns Pearl (2009). This susceptibility can undermine the effectiveness of SSL, particularly when dealing with diverse data environments.

Some works have been proposed to alleviate the effects of spurious correlations in SSL. Hamidieh et al. Hamidieh et al. (2024) introduced a method that counteracts these correlations by expanding the feature space, thereby providing more diverse training views to mitigate misleading associations. Park et al. Park et al. (2024) proposed that spuriously correlated attributes make neural networks inductively biased towards encoding lower effective rank representations and used rank regularization to eliminate biased samples. Another notable contribution comes from Chen et al.Zhu et al. (2023), who explored the use of a data reweighting strategy to reduce the importance of data samples that may contain spurious correlations. These methods attempt to eliminate spurious correlations by filtering or enhancing SSL samples at the sample level. Although this approach has proven effective—by excluding samples that may contain spurious correlations—it is difficult to ensure that the learned features are still reliable due to the partial unobservability of spurious correlations and variable coupling. In contrast, our work directly addresses the impact that spurious correlations might cause, utilizing the independence between unobserved variables and anchors under post-intervention distributions to ensure the reliability of the learned representations.

## E  TASK DISTRIBUTION & DATA DISTRIBUTION

**Task Distribution**: Task distribution refers to a set of tasks and their underlying distribution, where each task has its own specific objectives and associated data distribution. It is often used in meta-learning or multi-task learning scenarios to describe the diversity and variation across tasks.

For example, in a meta-learning scenario, the task distribution could include:

- A "cat vs dog" classification task (Task 1).
- A "car vs airplane" classification task (Task 2).
- A "bird vs fish" classification task (Task 3).

These tasks form the task distribution, and the meta-learning model is trained across this task space.

**Data Distribution**: Data distribution refers to the statistical distribution of data samples within a single task, typically described as the joint distribution $P(X, Y)$ of input $X$ and labels $Y$.

Task distribution describes the variability between tasks in a learning system, focusing on generalization across tasks. Data distribution focuses on the variability within a single task, addressing adaptation to specific data characteristics. The two concepts are hierarchical: task distribution governs the diversity of tasks, while each task has its own distinct data distribution.

**Reformulation of OOD generalization as generalization on task distributions**: We organize the whole process into the following steps:

**Step 1**: First, we provide the formal definition of task distribution.

Without loss of generality, let us use a classification task as an example. We define $X_{tr}^a = \{(x_i^a, y_i^a)\}_{i=1}^N$ as a training dataset, where $x_i^a$ represents a sample, $y_i^a$ represents the corresponding label, $a$ denotes the dataset index, and $N$ denotes the number of samples in the dataset. For a classification task, the goal is to learn a classifier $p^a(y_i^a|x_i^a)$, so that for any given sample $x_i^a$, the corresponding label can be predicted.

If $N \to +\infty$, $X_{tr}^a$ can be approximated as containing all the information necessary for the classification task and can thus be regarded as a complete dataset for a classification task. Simply put, the elements of a classification task include: the classifier and the dataset. We denote a task as $(X_{tr}^a, p^a(y_i^a|x_i^a))$. Then, the discrete distribution of tasks can be expressed as $\{X_{tr}^a, p^a(y_i^a|x_i^a)\}_{a=1}^M$, where $M$ represents the number of tasks.

Furthermore, when $a$ is different, the label space corresponding to $y_i^a$ is also different. For example, when $a = 1$, the label space is $\{Cat, Dog\}$, and when $a = 2$, the label space is $\{Plane, Train\}$. If $M \to +\infty$, $\{X_{tr}^a, p^a(y_i^a|x_i^a)\}_{a=1}^M$ can be regarded as a complete task distribution.

**Step 2**: Next, we reformulate SSL from the perspective of task distribution.

In Section 2 and Section 3.1, we explain why a mini-batch in SSL can be viewed as a task. Simply put, for a given mini-batch, it can be expressed as: $X_{tr,a}^{aug} = \{x_a^i, x_{anchor,a}^i\}_{i=1}^{2N}$, where $N$ denotes the number of ancestor samples in the mini-batch, $a$ represents the index of the mini-batch, and $x_{anchor,a}^i$ can be regarded as the label of the augmented sample. Meanwhile, the classifier to be learned for each mini-batch is modeled as $p^a(x_{anchor,a}^i|x_a^i)$.

Notably, the classifiers for all tasks in SSL are learned using the same classifier, i.e., the classifiers for all tasks aim to learn $p(x_{anchor,a}^i|x_a^i)$. For example, SimCLR models the classifier using a contrastive loss, while MAE models it using the $L_2$-norm. Therefore, whether D-SSL or G-SSL is used, as $M \to +\infty$, $\{(X_{tr,a}^{aug}, p(x_{anchor,a}^i|x_a^i))\}_{a=1}^M$ can be approximated as a task distribution, where $M$ represents the number of tasks.

**Step 3**: Finally, we reformulate the OOD generalization of SSL as generalization on task distributions.

In traditional machine learning, given training data, the goal is to learn $p(y|x)$. This can be understood as modeling the data distribution $p(x, y)$ as $p(x)p(y|x)$, where $p(y|x)$ is learned from the training data and transferred to the test data distribution $p(x)$. This approach assumes that the training and test data are identically and independently distributed, i.e., $p(x_{train}) = p(x_{test}) = p(x)$, and $p(x_{train}, y_{train}) = p(x_{test}, y_{test}) = p(x, y)$. Consequently, $p(x)p^{train}(y|x) = p(x)p^{test}(y|x)$, leading to $p^{train}(y|x) = p^{test}(y|x)$.

By analogy, when each data sample is treated as a task, the corresponding learning objective becomes $p(p^a(x_{anchor,a}^i|x_a^i)|X_{tr,a}^{aug})$. This learning goal is similar to that in meta-learning [1-2], where the goal is to learn a function that can output the classifier for a given task dataset. Therefore, when the training data are drawn from a task distribution, the learning objective is to model the task distribution, i.e., to learn $p(p^i(y|x)|p(\text{task } i))$, such that it applies to both training and test tasks. Since training and test tasks are different, from the perspective of the training tasks, the test tasks represent OOD scenarios. However, from the perspective of the task distribution, both training and test tasks belong to the same task distribution.

Thus, from the viewpoint of traditional machine learning, SSL can be considered as training with mini-batches of size 1, where each training sample is a training task. One open problem is how to model $p(p^i(y|x)|p(\text{task } i))$. Since we define the classifier $p(x_{anchor,a}^i|x_a^i)$ for each SSL training task as identical, $p(p^i(y|x)|p(\text{task } i))$ can be directly modeled as $p(x_{anchor,a}^i|x_a^i)$, which applies to any sample from any task.

In conclusion, combining **Step 1-3**, we reformulate OOD generalization as generalization on task distributions.

## F  Intuitive Explanations for **Assumption** 3.3 and **Theorem** 4.7

**Assumption** 3.3 illustrates that regardless of whether $e \in \mathcal{D}$ or $e \in$ PID, $x^+$ is generated under the control of two variables, $s$ and $x^{\text{label}}$. Therefore, given $x^+$, $s$ and $x^{\text{label}}$ are conditionally independent, regardless of the correlation between them.

From **Assumption** 3.3, the optimal $f$ should be $F_{x^{\text{label}}}$. However, without additional constraints, it is difficult to obtain this optimal $f$. **Theorem** 3.4 provides a way to obtain another good $f$, defined as $f^*$ in the theorem. Why is $f^*$ considered good? This is because **Theorem** 3.4 implies that when $\mathcal{D}$ is sufficiently large and diverse, an optimal $f^*$ trained on one distribution will perform worse than random guessing in some other environment. Under such conditions, no other $f$ obtained from training on any distribution can achieve better worst-case OOD performance than the PID. Why is focusing on the worst-case scenario better than other cases? During training, we minimize the worst-case scenario, which involves minimizing: $\max_{e \in \mathcal{D}} \mathcal{L}^e(p_f(x^{\text{label}}|x^+))..$ For any $f$, the term $\max_{e \in \mathcal{D}} \mathcal{L}^e(p_f(x^{\text{label}}|x^+))$ is always greater than or equal to $\mathcal{L}^e(p_f(x^{\text{label}}|x^+))$ for any specific environment $e$. If we learn an $f$ that minimizes the worst-case term $\max_{e \in \mathcal{D}} \mathcal{L}^e(p_f(x^{\text{label}}|x^+))$, then we naturally minimize $\mathcal{L}^e(p_f(x^{\text{label}}|x^+))$ for all $e$ in $\mathcal{D}$. This ensures robustness across all scenarios, making the worst-case optimization strategy effective for improving OOD performance.

The high-level explanation of **Theorem** 4.7 can be presented as follows: 1) From **Definition** 4.4, it follows that if $ba(s)$ can be identified, then $s$ and $x^{\text{label}}$ are conditionally independent given $ba(s)$; 2) In this paper, $ba(s)$ is implemented as described in Equation (5) in the main text. The key challenge lies in obtaining $s$. As shown in Section 4.1, we explain the identifiability of $s$, as well as how each label is modeled using a distribution for $s$. During implementation, we sample from this distribution to generate a series of discrete vectors that approximate $s$ associated with a specific label; 3) From Equation (2) in the main text, we have: $p(x^+, x^{\text{label}}, s) = p(x^+|x^{\text{label}}, s)p(x^{\text{label}})p(s|x^{\text{label}})$. If we select sample pairs for a mini-batch such that all pairs share the same $ba(s)$, the resulting mini-batch can be considered as constructed under the same $ba(s)$. In other words, the samples in the mini-batch are conditioned on $ba(s)$. Combined with the argument in **Point 1**), we have: $p(x^+, x^{\text{label}}, s) = p(x^+|x^{\text{label}}, s)p(x^{\text{label}})p(s)$, which ensures that the mini-batch satisfies PID. The key to achieving PID is ensuring that all sampled examples in the mini-batch have consistent $ba(s)$, i.e., the background information is the same. This allows SSL training to focus on foreground information while disregarding background information.

## G  More Explanation for the Identifiability of Spurious Variable

To better address the identifiability of spurious variables in the context of SSL, we organize the response into the following steps:

**Step 1**: First, we need to clarify that in Section 2 and Section 3.1, we propose a new perspective for understanding SSL. Taking classification as an example, under this new perspective, each mini-batch during the training phase of SSL can be treated as an independent multi-classification task. Different mini-batches correspond to different classification tasks. In contrast, the traditional perspective of SSL considers the entire dataset as a single task for unsupervised learning.

Therefore, under the new perspective, the training samples in each mini-batch can be considered labeled. Whether these labels are accurate is not our concern for now, as this falls under the domain of Bayesian error. Consequently, in this sense, spurious variables can be identifiable.

**Step 2**: We first explain what we mean by the distribution of tasks, using classification as an example. A learning task can be narrowly defined as assigning a label to each sample in a dataset, where the label types are finite. This dataset can represent the task, and the entirety of the dataset can be regarded as the data distribution of that task. Thus, different tasks correspond to different datasets, with distinct label types (tasks with the same label types are considered the same). In this way, a task distribution is essentially the distribution over these datasets, with each element of the task distribution corresponding to a specific dataset.

**Step 3**: Next, we point out that in this paper, the spurious variable $s$ indeed takes values in an infinite space since it is represented by a high-dimensional vector. The values of this vector can be arbitrary.

We must define $s$ as taking infinite values because, as discussed in Section 2 and Section 3.1, we reinterpret SSL as learning a task distribution where the label types involved are infinite.

Different labels may correspond to different latent variables. These differences are represented by different distributions, i.e., we model the distribution $q_\phi(s|x^+, x^{\mathrm{label}})$ using a latent variable model. This allows us to derive the distribution of the spurious variable $s$ for any given label. The values of the probability density can be understood as the degree of correlation between a specific label and a particular value of the latent variable. Hence, given a label, once its conditional distribution $q_\phi(s|x^+, x^{\mathrm{label}})$ is determined, we can estimate the corresponding spurious variable $s$ through sampling.

**Step 4**: We do not theoretically prove that the latent variable model can directly identify the spurious variable $s$. In this paper, the identification of $s$ is based on a strong assumption—**Assumption** 4.1 in the paper. This assumption is justified as follows:

Based on the literature (Blei et al., 2017; Sriperumbudur et al., 2013), which expresses the true prior in closed form, we deduce that when the causal relationship between the latent covariate and the label changes with the tasks, an exponential family distribution is capable of modeling the conditional distribution $p(s|x^{\mathrm{label}})$.

Combining **Step 1**, **Step 2**, and **Step 3**, we satisfy the condition that the causal relationship between the latent covariate and the label changes with the tasks.

**Step 5**: **Theorem** 4.3 is also based on **Assumption** 4.1. The key result of **Theorem** 4.3 is that we can uniquely identify $\phi$ and $(\mathrm{f}, g, \mathrm{A})$. However, this strong assumption imposes certain limitations on the accuracy of spurious variable identification, which is a topic for future research. Despite this strong assumption, our experimental results demonstrate the effectiveness of our method.

