# OpenReview forum: "On the Out-of-Distribution Generalization of Self-Supervised Learning"
_ICLR.cc/2025/Conference — Submitted to ICLR 2025_

### Official Review · Reviewer_fPhX · 2024-10-29

**Soundness:** 3
**Presentation:** 2
**Contribution:** 3
**Rating:** 6
**Confidence:** 3

**Summary:**

This paper regards the mini-batches in the SSL training as the environments (domains) in OOD generalization problems and proposes that each mini-batch can be viewed as a multi-class classification task. Based on this formulation, the authors points out that when the similarity is measured using non-causal features, SSL will learn spurious representations. To address this issue, the authors propose to model the Post-Intervention Distribution (PID) using VAEs for each mini-batch and further propose a mini-batch sampling strategy that selects samples with similar balancing scores based on the  $p^e(s|x^{label})$ learned by the VAE. The experiments demonstrat the effectiveness of the method.

**Strengths:**

1. The perspective of converting the SSL training to a domain-generalization-like problem using an SCM is natrual and interesting.
2. The proposed method is built with theoretical guarantees on the identifiability of the distribution parameters and the recover of the PID.
3. The experiments cover many scenarios, including semi-supervised learning, transfer learning, and few-shot learning tasks. The improvements of the proposed method are significant.

**Weaknesses:**

1. Some points are not quite clear and need further clarification. For example, despite Theorem 4.7, it is a bit confusing that why sampling samples with the same propensity score would help to recover $p^{PI}$. It would be better to provide some high-level explanations.
2. The authors didn't evaluate their method on classic OOD tasks like PACS, OfficeHome, ColoredMNIST, etc. Since this work aims to improve SSL's OOD performance, it would be necessary to evaluate these tasks. Otherwise, the author should explain why not doing so.

**Questions:**

1. Could you shed light on why using an exponential family distribution to model $p(s|x^{label})$?
2. In line 227, how does Theorem 3.4 "implies that when $\mathcal{D}$ is sufficiently large and diverse, an optimal $f^*$ trained on one distribution will perform worse than random guessing in some other environment."?
3. In line 459, why "We can observe that the performance of BYOL rapidly deteriorates with batch size."? It seems that BYOL suffers from smaller performance degradation than BYOL+ours.
4. In line 267, should it be $T_{ij}=a_{ij}\times \cdot$?

---

> ### Author Response · Authors · 2024-11-21
> **Response to Weaknesses 1**
>
> Thank you for pointing this out. We provide a high-level explanation of **Theorem 4.7** as follows:
>
> ---
>
> 1) From **Definition 4.4**, it follows that if $ ba(s) $ can be identified, then $ s $ and $ x^{\rm label} $ are conditionally independent given $ ba(s) $.
>
> 2) In this paper, $ ba(s) $ is implemented as described in Equation (5) in the main text. The key challenge lies in obtaining $ s $. In our response to **Reviewer 1's first weakness**, we explained the identifiability of $ s $, as well as how each label is modeled using a distribution for $ s $. During implementation, we sample from this distribution to generate a series of discrete vectors that approximate $ s $ associated with a specific label.
>
> 3) From **Equation (2)** in the main text, we have: $
>    p(x^+, x^{\rm{label}}, s) = p(x^+ | x^{\rm{label}}, s) p(x^{\rm{label}}) p(s | x^{\rm{label}})
>    $. If we select sample pairs for a mini-batch such that all pairs share the same $ ba(s) $, the resulting mini-batch can be considered as constructed under the same $ ba(s) $. In other words, the samples in the mini-batch are conditioned on $ ba(s) $. Combined with the argument in **Point 1**, we have: $
>    p(x^+, x^{\rm{label}}, s) = p(x^+ | x^{\rm{label}}, s) p(x^{\rm{label}}) p(s)
>    $, which ensures that the mini-batch satisfies PID.
>
> ---
>
> The key to achieving PID is ensuring that all sampled examples in the mini-batch have consistent $ ba(s) $, i.e., the background information is the same. This allows SSL training to focus on foreground information while disregarding background information.

---

> ### Author Response · Authors · 2024-11-21
> **Response to Weaknesses 2**
>
> Thank you for pointing this out.
>
> ---
>
> Following the reviewer's suggestions, we constructed three sets of toy experiments to evaluate the effectiveness of PID on OOD tasks, including the PACS, OfficeHome, and ColoredMNIST datasets. Specifically, we assessed the performance of SSL baselines on these three datasets before and after incorporating the proposed PID.
>
> ---
>
> PACS:
>
> | Model       | Photo | Sketch | Cartoon | Painting (Unseen) | Average |
> | ----------- | ----- | ------ | ------- | ----------------- | ------- |
> | SimCLR      | 86.4  | 85.1   | 87.2    | 74.3              | 80.7    |
> | SimCLR+Ours | 88.0  | 87.4   | 90.1    | 79.2              | 85.0    |
> | BYOL        | 83.9  | 84.6   | 82.7    | 64.5              | 74.2    |
> | BYOL+Ours   | 84.2  | 86.9   | 85.0    | 70.8              | 78.9    |
>
> ---
>
> OfficeHome:
>
> | Method      | A → C | A → P | A → R | C → A | C → P | C → R | P → A | P → C | P → R | R → A | R → C | R → P |
> | ----------- | ----- | ----- | ----- | ----- | ----- | ----- | ----- | ----- | ----- | ----- | ----- | ----- |
> | SimCLR      | 58.2  | 63.5  | 69.8  | 78.9  | 69.7  | 66.8  | 63.4  | 52.3  | 58.4  | 56.1  | 72.9  | 71.0  |
> | SimCLR+Ours | 61.1  | 65.2  | 71.9  | 81.1  | 72.0  | 68.2  | 67.5  | 59.1  | 59.9  | 61.2  | 74.8  | 73.5  |
>
> ---
>
> ColoredMNIST:
>
> | Method      | Accuracy |
> | ----------- | -------- |
> | SimCLR      | 85.2     |
> | SimCLR+Ours | 88.6     |
>
> The results demonstrate that our method consistently improves performance, confirming its effectiveness on OOD tasks.
>
> ---
>
> Secondly, we would like to clarify that we have provided an evaluation of PID on OOD tasks in Section 5.2 (L413-426). Specifically, Tables 3 and 4 present the transfer learning performance of the proposed method under both standard and few-shot settings. The pre-training and test datasets are based on different benchmarks. The results show that our method consistently enhances performance across all SSL baselines, demonstrating its effectiveness on OOD tasks.

---

> ### Author Response · Authors · 2024-11-21
> **Response to Question 1**
>
> Thank you for pointing this out. The rationale for this approach is based on the findings in the literature [1, 2], which provide a closed-form expression for the true prior. These studies demonstrate that when the causal relationship between the latent covariate and the label varies across tasks, an exponential family distribution can effectively model the conditional distribution $ p(s|x^{\rm{label}}) $.
>
> Combining **Step 1** and **Step 2** from the **Response to Weakness 1** addressed to **Reviewer 1**, we precisely arrive at the conclusion that the causal relationship between the latent covariate and the label changes with the tasks.
>
> [1] David M. Blei, Alp Kucukelbir, and Jon D. McAuliffe. Variational inference: A review for statisticians. Journal of the American Statistical Association, pp. 859–877, Apr 2017. doi:10.1080/01621459.2017.1285773.
>
> [2] BharathK. Sriperumbudur, Kenji Fukumizu, Arthur Gretton, Aapo Hyv¨arinen, and Revant Kumar. Density estimation in infinite dimensional exponential families. arXiv: Statistics Theory,arXiv: Statistics Theory, Dec 2013.

---

> ### Author Response · Authors · 2024-11-21
> **Response to Question 2**
>
> Thank you for pointing it out. **Assumption 3.3** illustrates that regardless of whether $ e \in \mathcal{D} $ or $ e \in \text{PID} $, $ x^+ $ is generated under the control of two variables: $ s $ and $ x^{\rm label} $. Therefore, given $ x^+ $, $ s $ and $ x^{\rm label} $ are conditionally independent, regardless of the correlation between them.
>
> From **Assumption 3.3**, the optimal $ f $ should be $ F_{x^{\rm label}} $. However, without additional constraints, it is challenging to obtain this optimal $ f $. **Theorem 3.4** provides a pathway to identify another good $ f $, denoted as $ f^* $.
>
> ---
>
> ### Why is $ f^* $ good?
>
> **Theorem 3.4** implies that when $ \mathcal{D} $ is sufficiently large and diverse, an optimal $ f^* $ trained on one distribution will perform worse than random guessing in some other environments. Under such conditions, no other $ f $ obtained from training on any distribution can achieve better worst-case OOD (Out-Of-Distribution) performance than the PID.
>
> ---
>
> ### Why is the worst-case scenario better than others?
>
> During training, we minimize the loss in the worst-case scenario: $
> {\max _{e \in \mathcal{D}}} {\mathcal{L}}^e({p_f}(x^{\rm{label}} | x^+))
> $. For any $ f $, the worst-case loss $
> {\max _{e \in \mathcal{D}}} {\mathcal{L}}^e({p_f}(x^{\rm{label}} | x^+))
> $ is always greater than or equal to the loss in any specific environment \( e \): $
> {\mathcal{L}}^e({p_f}(x^{\rm{label}} | x^+)).
> $
>
> If we learn an $ f $ that minimizes the worst-case loss $
> {\max _{e \in \mathcal{D}}} {\mathcal{L}}^e({p_f}(x^{\rm{label}} | x^+))
> $, then $ f $ naturally minimizes $
> {\mathcal{L}}^e({p_f}(x^{\rm{label}} | x^+))
> $ for all $ e $ in $ \mathcal{D} $. This ensures robust performance across all scenarios, making the worst-case optimization strategy particularly effective for enhancing OOD generalization.

---

> ### Author Response · Authors · 2024-11-21
> **Response to Question 3**
>
> Thank you for pointing it out. We apologize for the incorrect labels in **Figure 4**. In **Figure 4**, the blue line should represent **BYOL + ours**, and the red line should represent **BYOL**.

---

> ### Author Response · Authors · 2024-11-21
> **Response to Question 4**
>
> Thank you for pointing it out. Indeed, **line 267** should be $ {\rm T}_{ij}( \cdot ) = $. We apologize for this error.

---

> ### Author Response · Authors · 2024-11-23
> **Response to Reviewer fPhX**
>
> First, we sincerely appreciate for your feedback, which has greatly encouraged us. Then, we provide more explanation for **Question 2**.
>
> ---
>
> **Re-Response to Question 2**
>
> In fact, **Theorem 3.4** is not intended to illustrate that an optimal $f^* $ trained on one distribution will perform worse than random guessing in some other environment. Rather, its purpose is to explain how we can obtain a surrogate when the optimal $ f^* $ is not accessible, and how to prove that this surrogate is effective.
>
> From **Assumption 3.3**, the optimal $ f^* $ should be $ F_{x^{\rm label}} $. However, without additional constraints, it is challenging to obtain this optimal $ f^* $.
>
> In **Response to Question 2**, our goal is to explain what the surrogate is and why it is effective.
>
> ---
>
> We would like to express our gratitude again for your recognition of our work and for the time and effort you have dedicated to reviewing it.

---

### Official Review · Reviewer_priS · 2024-11-03

**Soundness:** 3
**Presentation:** 3
**Contribution:** 3
**Rating:** 6
**Confidence:** 3

**Summary:**

This paper introduces a training batch sampling strategy designed to enhance self-supervised learning and improve generalization beyond the training distribution. The approach is inspired by the concept of invariant causal structure across different environments: while causal relationships between features and labels remain consistent, spurious correlations vary across environments. The proposed methodology employs a constraint, PID, during mini-batch sampling, which disregards spurious correlations and supports out-of-distribution generalization.

**Strengths:**

The main strength of the paper lies

**Weaknesses:**

At times, the presentation is overly technical or abstract, which might be challenging for practitioners who seek to grasp the main insights of the paper. The core message is to introduce a sampling strategy combined with a distributional constraint (PID) that encourages the self-supervised method to disregard correlations that change across domains and focus on stable, causal correlations. The objective is to enhance out-of-distribution generalization by learning these invariant structures. Adding a non-technical explanation, perhaps as a remark, on how the algorithm achieves PID enforcement would be beneficial. Please refer to my questions below for further clarification.

**Questions:**

There are also a few concerns/typos that need to be taken care of for better readability:

1. In Algorithm 1, the steps, especially the count of i, seem to be a bit confusing. It may be better to write: "Set $i \leftarrow 0$", and select the initial pair $(x_0^+, x_0^{\rm label})$. Then, for $i \ge 1$, write the two steps and finally add, "Set $i \leftarrow i + 1$".

2. The number of samples mu should be $\mu$, I guess?

3. It seems that the definition of PID is the same as assuming $x^{\rm label}$ and $s$ are independent. Maybe it would be easier to present it that way.

4. What is $\mathcal{L}^{\rm PID}$? Is it $\mathcal{L}^{\rm e}, e \in \mathcal{D}$ where $e$ satisfies PID? How is $f$ related to $F$ in Equation (1)? Is $f$ a generic function in the class of hypothesis and $F$ the true generating function?

5. Are we assuming that minimizer $f^*$ is same for all distributions in $\mathcal{D}$ that satisfies PID?

6. I am a little bit confused about Assumption 3.3. In PID, we have $x^{\rm label}$ is independent of $s$, whereas in Assumption 3.3, we also have $x^{\rm label}$ is independent of $s$ given $x^+$. Are we assuming Assumption 3.3 for all distributions in $\mathcal{D}$? A remark with some intuitive explanation of Theorem 3.4 would be helpful.

---

> ### Author Response · Authors · 2024-11-21
> **Response to Question 1**
>
> Thank you for your advice. In the revised version, we have updated **Algorithm 1** accordingly.

---

> ### Author Response · Authors · 2024-11-21
> **Response to Question 2**
>
> Thank you for pointing this out. The definition of $ nu $ is provided in **Line 313** of the original submission, and the definition of $ mu $ is provided in **Line 333** of the original submission.

---

> ### Author Response · Authors · 2024-11-21
> **Response to Question 3**
>
> Thank you for pointing this out. To further clarify PID, in the revised version, we have added this explanation to the definition of PID.

---

> ### Author Response · Authors · 2024-11-21
> **Response to Question 4**
>
> Thank you for pointing these out.
>
> $ {\mathcal{L}}^{\rm PID} $ represents the loss computed on a dataset satisfying the PID constraints and can also be understood as $ {\mathcal{L}}^{e} $, where $ e \in \text{PID} $. According to **Equation (1)**, $ F $ is a function that generates $ x^+ $ based on $ x^{\rm label} $. From **Assumption 3.3**, we can deduce that the optimal $ f $ should be $ F_{x^{\rm label}} $. However, without additional constraints, it is challenging to obtain this optimal $ f $.
>
> **Theorem 3.4** provides a pathway to obtain another good $ f $, defined as $ f^* $ in the theorem.
>
> Why is $ f^* $ considered good? This is because **Theorem 3.4** implies that when $ \mathcal{D} $ is sufficiently large and diverse, an optimal $ f^* $ trained on one distribution will perform worse than random guessing in some other environments. Under such conditions, no other $ f $ obtained from training on any distribution can achieve better worst-case OOD performance than that trained under the PID constraints.

---

> ### Author Response · Authors · 2024-11-21
> **Response to Question 5**
>
> Thank you for pointing this out. Indeed, we assume that the minimizer $ f^* $ is the same for all distributions in $ \mathcal{D} $ that satisfy the PID constraints. This assumption underscores the importance of PID.
>
> In our **Response to Question 4**, we mentioned the worst-case scenario. Here, we further explain why focusing on the worst-case is beneficial compared to other scenarios. During training, we minimize the worst-case scenario, which means we minimize: ${\max _{e \in \mathcal{D}}} {\mathcal{L}}^e({p_f}(x^{\rm{label}} | x^+))$.
>
> For any $ f $, the term ${\max _{e \in \mathcal{D}}} {\mathcal{L}}^e({p_f}(x^{\rm{label}} | x^+))$ is always greater than or equal to ${\mathcal{L}}^e({p_f}(x^{\rm{label}} | x^+))$  for any specific environment $ e $.
>
> By learning an $ f $ that minimizes the worst-case term ${\max _{e \in \mathcal{D}}} {\mathcal{L}}^e({p_f}(x^{\rm{label}} | x^+))$, we naturally achieve minimization of  ${\mathcal{L}}^e({p_f}(x^{\rm{label}} | x^+))$ for any environment $ e $ within $ \mathcal{D} $. This ensures robust performance across all scenarios, making worst-case optimization a strong strategy for improving generalization under diverse conditions.

---

> ### Author Response · Authors · 2024-11-21
> **Response to Question 6**
>
> Thank you for pointing these out. **Assumption 3.3** illustrates that regardless of whether $ e \in \mathcal{D} $ or $ e \in \text{PID} $, $ x^+ $ is generated under the control of two variables, $ s $ and $ x^{\rm label} $. Therefore, given $ x^+ $, $ s $ and $ x^{\rm label} $ are conditionally independent, regardless of the correlation between them.
>
> From **Assumption 3.3**, the optimal $ f $ should be $ F_{x^{\rm label}} $. However, without additional constraints, it is difficult to obtain this optimal $ f $. **Theorem 3.4** provides a way to obtain another good $ f $, defined as $ f^* $ in the theorem.
>
> ---
>
> Why is $ f^* $ considered good? This is because **Theorem 3.4** implies that when $ \mathcal{D} $ is sufficiently large and diverse, an optimal $ f^* $ trained on one distribution will perform worse than random guessing in some other environment. Under such conditions, no other $ f $ obtained from training on any distribution can achieve better worst-case OOD (Out-Of-Distribution) performance than the PID.
>
> ---
>
> Why is focusing on the worst-case scenario better than other cases? During training, we minimize the worst-case scenario, which involves minimizing:  $
> {\max _{e \in \mathcal{D}}} {\mathcal{L}}^e({p_f}(x^{\rm label} | x^+)).
> $. For any $ f $, the term  $
> {\max _{e \in \mathcal{D}}} {\mathcal{L}}^e({p_f}(x^{\rm label} | x^+))
> $ is always greater than or equal to $
> {\mathcal{L}}^e({p_f}(x^{\rm label} | x^+))
> $ for any specific environment $ e $.
>
> If we learn an $ f $ that minimizes the worst-case term $
> {\max _{e \in \mathcal{D}}} {\mathcal{L}}^e({p_f}(x^{\rm label} | x^+))
> $, then we naturally minimize $
> {\mathcal{L}}^e({p_f}(x^{\rm label} | x^+))
> $ for all $ e $ in $ \mathcal{D} $. This ensures robustness across all scenarios, making the worst-case optimization strategy particularly effective for improving OOD performance.

---

### Official Review · Reviewer_M58z · 2024-11-05

**Soundness:** 2
**Presentation:** 2
**Contribution:** 2
**Rating:** 5
**Confidence:** 4

**Summary:**

This paper inspects SSL from a causal perspective, which assumes a SCM for generating augmentations in both generative and discriminative approaches. To address spurious correlations between images and their non-semantic features, e.g., backgrounds and styles, the paper proposes rebalancing the training batches by sampling to decorrelate images from their non-semantic features.  Experiments show enhanced performances across various existing SSL methods.

**Strengths:**

1. The proposed rebalancing technique can be embedded into general SSL procedures, whether discriminative or generative, allowing for wide applicability.

2. Experiments are extensive in scope, covering both discriminative and generative SSL (appendix). Multiple learning tasks under distribution shift is considered, including semi-supervised, transfer learning and few-shot learning. A clear improvement of around 3% in accuracy is reported for most results.

**Weaknesses:**

In general, I am not convinced that the proposed SCM for generating augmentation, especially the characterization of spurious correlation, is relevant for OOD generalization of SSL.

1. The proposed SCM and the rebalancing strategy does not address the identifiability of spurious variables in the context of SSL. Spurious variables $s$ in supervised machine learning are the variables rejected by the conditional independence test $Y \perp e | s$, where $e$ is the group label. However, spurious variables are generally not identifiable without labels.  Literature has introduced inductive bias, e.g., simplicity bias, to identify spurious variables for SSL [1]. However, the SCM in the paper does not consider similar assumptions to address the identifiability of $s$. For example, in Figure 1(b), $s$ and $x^{label}$ (raw image) hold symmetric roles in the SCM. Since $s$ is learned as a latent variable by variational inference, there can be infinitely many solutions of $s$.  The identifiability results in Theorem 4.3 does not resolve the identifiablity of $s$, because it depends on the condition that $p(x^+|x^{label},s)$ is learned, implying $s$ has been identified.
2. The conditional independence implied by the SCMs may not reflect practice in SSL.  The PID SCM in Fig.3 models the statistical independence between styles or backgrounds (s) and images ($X^{label}$), but the style and background can be directly identified from the image in practice. In general, $s$ is always measurable with respect to $X^{label}$. Similarly, both $X^{label}$ and $s$ are direct causes of $X^{+}$ in Fig.2, which is also inconsistent to the augmentation practice that takes as input the raw images only, since the background is just part of the raw image. Does this paper consider a special augmentation procedure?
3. I identify a gap between self-supervised representation learning, whose target is $p(X^{label})$, and the models used in theory. The binary classification model in Proposition 3.1 learns the density ratio $p(X^{label})/p(X^{+})$, and the "alignment" model in Theroem 3.4 learns $p(X^{label}|p^{+})$. The paper has not addressed that a non-spurious classification model or "alignment" model implies a non-spurious generative model. A simple counterexample: assume that the augmentation procedure retains the style of the image. The classifier does not depend on the style to distinguish between anchor and augmentations because they share the same style. However, styles can still be learned by the generative model.

Moreover, I think this paper can be substantially improved in writing for its message to be more effectively conveyed.

4. Some concepts and statements are not well defined and formulated.
   - The "task distribution" is not defined. In L131-132, a statement is made that "this framework involves estimating the true task distribution from discrete training tasks, enabling the SSL model to generalize to new, unseen tasks (i.e., test tasks)." Is task modeled as a random variable? What does task correspond to in the SCM? For example, if task refers to batch index in Fig.2(b), then generalization is essentially impossible because training and test batch indices do not overlap. If task refers to batches of X in Fig.2(b), than generalization is only possible when the image batches are i.i.d., which is irrelevant to OOD generalization. In L157, the author states that s, denoting the style or background, does not contain any causal semantics related to the task. This statement contradicts the SCM in Fig 1 as well, where s is a direct cause of X+. Therefore, the definition of "task" is more vague here.
   - What does the statement mean that "$x^{label}$ is regarded as the label" (L245), since $x^{label}$ is the raw image? A formulation of this equivalence may help improve clarity.
5. Models, assumptions and theorem statements are not explicitly presented.
   - I understand the benefits for deferring formal theorem statements to the appendix. However, the formal statement of Proposition 3.1 is missing in both the main text and the appendix. The assumption of mixture of gaussians, balanced labels, equal dimensions between spurious variables and images, and the model of binary classification are all woven into the proof.
   - This paper models the SSL procedure by two parts: a classification model and a conditional generative alignment model. The formulation of the classification model is mixed in the proof.  The alignment model is not formulated until Theorem 3.4. However, since the learning procedure is repeatedly mentioned throughout the theory, I suggest a clear statement of the models at the beginning.
6. Multiple notations are unexplained.
   - The notation  $L^{PID}$ in L225 is vague because PID is a family of distributions. Which distribution is the loss evaluated with respect to?
   - Similarly, $\perp_{PI}$ in L217 is also unexplained. Is the independence condition satisfied for all PI distributions?
   - nu in L313 and mu in L315, 333.
7. The implication of the identifiability result in Theorem 4.3 is insufficiently addressed. Also related to the first point, what does the equivalance in Definition 4.2 imply for the identiability of spurious variables, and more importantly, the generative model?

Minor points:

8. Experiments are in relatively small scale. The results are presented for Imagenet-100 instead of the more popular ImageNet-1k. Models are trained with a limited number of epochs.
9. There has been theoretical and empirical analysis of the vulnerability of SSL to spurious correlation, e.g., in [1]. Related work on spurious correlation in SSL can be reviewed to establish the paper's position in the broader literature.

[1] Hamidieh, K., Zhang, H., Sankaranarayanan, S., & Ghassemi, M. (2024). Views Can Be Deceiving: Improved SSL Through Feature Space Augmentation.

**Questions:**

1. Why does $f^\star$ maximize the loss function in L225, since the proof indicates minimization instead?

---

> ### Author Response · Authors · 2024-11-21
> **Response to Weaknesses 1**
>
> To better address the identifiability of spurious variables in the context of SSL, we organize the response into the following steps:
>
> ---
>
> ### Step 1:
>
> First, we need to clarify that in **Section 2**, we propose a new perspective for understanding SSL. Taking classification as an example, under this new perspective, each mini-batch during the training phase of SSL can be treated as an independent multi-classification task. Different mini-batches correspond to different classification tasks. In contrast, the traditional perspective of SSL considers the entire dataset as a single task for unsupervised learning.
>
> Therefore, under the new perspective, the training samples in each mini-batch can be considered labeled. Whether these labels are accurate is not our concern for now, as this falls under the domain of Bayesian error. Consequently, in this sense, spurious variables can be identifiable.
>
> ---
>
> ### Step 2:
>
> #### 2.1 Definition of Task Distribution
>
> We first explain what we mean by the distribution of tasks, using classification as an example. A learning task can be narrowly defined as assigning a label to each sample in a dataset, where the label types are finite. This dataset can represent the task, and the entirety of the dataset can be regarded as the data distribution of that task. Thus, different tasks correspond to different datasets, with distinct label types (tasks with the same label types are considered the same). In this way, a task distribution is essentially the distribution over these datasets, with each element of the task distribution corresponding to a specific dataset.
>
> #### 2.2 Infinite Nature of Spurious Variable $ s $
>
> Next, we point out that in our **submission**, the spurious variable $ s $ indeed takes values in an infinite space since it is represented by a high-dimensional vector. The values of this vector can be arbitrary. We must define $ s $ as taking infinite values because, as discussed in **Section 2** and the first paragraph of **Section 3.1**, we reinterpret SSL as learning a task distribution where the label types involved are infinite.
>
> Different labels may correspond to different latent variables. These differences are represented by different distributions, i.e., we model the distribution $q_{\phi}(s|x^+,x^{\rm{label}})$ using a latent variable model. This allows us to derive the distribution of the spurious variable $ s $ for any given label. The values of the probability density can be understood as the degree of correlation between a specific label and a particular value of the latent variable. Hence, given a label, once its conditional distribution $q_{\phi}(s|x^+,x^{\rm{label}})$  is determined, we can estimate the corresponding spurious variable $ s $ through sampling.
>
> ---
>
> ### Step 3:
>
> We do not theoretically prove that the latent variable model can directly identify the spurious variable $ s $. In our **submission**, the identification of $ s $ is based on a strong assumption—**Assumption 4.1** in the paper. This assumption is justified as follows:
>
> Based on the literature [1, 2], which expresses the true prior in closed form, we deduce that when the causal relationship between the latent covariate and the label changes with the tasks, an exponential family distribution is capable of modeling the conditional distribution $ p(s|x^{\rm{label}}) $.
>
> Combining **Step 1** and **Step 2**, we satisfy the condition that the causal relationship between the latent covariate and the label changes with the tasks.
>
> ---
>
> ### Step 4:
>
> **Theorem 4.3** is also based on **Assumption 4.1**. The key result of Theorem 4.3 is that we can uniquely identify $ \phi $ and $ ({\rm{f}},{g},{\rm{A}}) $. However, this strong assumption imposes certain limitations on the accuracy of spurious variable identification, which is a topic for future research. Despite this strong assumption, our experimental results demonstrate the effectiveness of our method.
>
> ---
>
> [1] David M. Blei, Alp Kucukelbir, and Jon D. McAuliffe. Variational inference: A review for statisticians. Journal of the American Statistical Association, pp. 859–877, Apr 2017. doi:10.1080/01621459.2017.1285773.
>
> [2] BharathK. Sriperumbudur, Kenji Fukumizu, Arthur Gretton, Aapo Hyv¨arinen, and Revant Kumar. Density estimation in infinite dimensional exponential families. arXiv: Statistics Theory,arXiv: Statistics Theory, Dec 2013.

---

> ### Author Response · Authors · 2024-11-21
> **Response to Weaknesses 2**
>
> Thank you for pointing these out. This paper aims to propose a sampling method to construct mini-batches that satisfy PID (Pairwise Independence and Diversity), a process independent of data augmentation. The basic idea of constructing PID can be summarized as follows:
>
> ---
>
> 1) **From Definition 4.4**, as long as $ ba(s) $ is identified, $ s $ and $ x^{\rm{label}} $ are conditionally independent given $ ba(s) $.
>
> 2) In this paper, $ ba(s) $ can be implemented through Equation (5) in the main text. The critical aspect here is how to obtain $ s $. In our response to the first reviewer question, we explained the identifiability of $ s $, as well as how each label is modeled with a corresponding distribution for $ s $. During the actual implementation, we sample from this distribution to obtain a series of discrete vectors that approximate $ s $ associated with a specific label.
>
> 3) From Equation (2) in the main text, we have:  $p(x^+, x^{\rm{label}}, s) = p(x^+|x^{\rm{label}}, s) p(x^{\rm{label}}) p(s|x^{\rm{label}})$. If we select sample pairs for a mini-batch such that all pairs have the same $ ba(s) $, the resulting mini-batch can be considered constructed under the same $ ba(s) $. In other words, the samples in the mini-batch can be regarded as conditioned on $ ba(s) $. Combined with the first conclusion (refer to **Point 1.**), we have: $p(x^+, x^{\rm{label}}, s) = p(x^+|x^{\rm{label}}, s)p(x^{\rm{label}})p(s)$. This means the constructed mini-batch satisfies PID.
>
> ---
>
> The key to achieving PID is ensuring that all sampled examples share the same $ ba(s) $, i.e., the background information is consistent. This ensures that during training, SSL focuses on the foreground information while discarding the background information.

---

> ### Author Response · Authors · 2024-11-21
> **Response to Weaknesses 3**
>
> Thank you for pointing these out. Based on **Section 2** and **Section 3.1**, under the new perspective, both D-SSL (Discriminative SSL) and G-SSL (Generative SSL) share a common learning objective: aligning the positive sample in a pair with its corresponding anchor. Thus, the learning objectives of D-SSL and G-SSL can be unified as maximizing $ p(x^{\rm{label}} | x^+) $. The difference lies in how they achieve $ p(x^{\rm{label}} | x^+) $: for example, contrastive learning uses a contrastive loss, while MAE employs the $ L_2 $-norm.
>
> ---
>
> Secondly, our argument is that, in general, both D-SSL and G-SSL tend to encode task-irrelevant information into feature representations during the learning process. However, D-SSL and G-SSL trained with mini-batches that satisfy the PID constraints can mitigate this challenge. The reason for this is provided in **Theorem 3.4**. Simply put, **Theorem 3.4** implies that when $ \mathcal{D} $ is sufficiently large and diverse, an optimal $ f^* $ trained on one distribution will perform worse than random guessing in some other environments. Under such conditions, no other $ f $ trained on any distribution can achieve better worst-case OOD performance than PID. This conclusion motivates us to design a new batch sampling strategy to ensure that the resulting batches satisfy the PID constraints, thereby improving the OOD generalization of SSL models.
>
> ---
>
> Finally, we provide an intuitive explanation of why G-SSL benefits from PID. From the above response to the second reviewer question, we know that the key to achieving PID lies in ensuring that all sampled examples in a mini-batch have the same $ ba(s) $, i.e., the background information is consistent. Since the core idea of G-SSL is to reconstruct masked sub-regions of images, and the background across the mini-batch is similar, the difficulty of reconstructing the background is significantly lower than reconstructing varying foregrounds. This, to some extent, constrains G-SSL to focus more on foreground-related semantic information when learning feature representations.

---

> ### Author Response · Authors · 2024-11-21
> **Response to Weaknesses 4.1**
>
> First, let us explain what is meant by a task distribution, using a classification problem as an example. A learning task can be narrowly defined as assigning a label to each sample in a dataset, with a finite number of label types. This dataset can represent the task, and all the data in the dataset can be regarded as the data distribution for that task. Different tasks correspond to different datasets, with distinct label types (if the label types are the same, the tasks are considered the same). In this way, the task distribution is essentially the distribution of these datasets, where each element in the task distribution corresponds to a specific dataset.
>
> ---
>
> Secondly, in this paper, the **SCM** (Structural Causal Model) models the learning process for each specific task, specifically by aligning the positive sample in a pair with its corresponding anchor. **Figure 2(a)** illustrates that during the training phase, there exist some training tasks where the background information of images may vary across different categories, leading to the model learning background information. **Figure 2(a)**  and **Figure 2(b)** together demonstrate that the structure of the SCM varies across tasks, making it challenging to model every task with a unified SCM structure.
>
> ---
>
> Finally, we provide an intuitive explanation for the OOD generalization ability of SSL. Based on **Section 2** and **Section 3.1**, each mini-batch in the SSL training process can be regarded as a training task. The SSL training process can thus be viewed as task-based training. The entire training process can be likened to training with mini-batches of size one, where each sample represents a distinct task.
>
> Under the assumption that the training data or tasks are sufficiently large, compared to the traditional machine learning process, which can be viewed as modeling data distributions, the SSL training process models task distributions. Once SSL successfully models the task distribution, it can perform well on any sample (i.e., specific task) within the distribution, thus exhibiting OOD capability (when test tasks differ from training tasks).
>
> It is important to note why the training process in traditional machine learning cannot be considered task-based. This is because, in traditional machine learning, every mini-batch in training represents the same task, i.e., the label space remains consistent. In contrast, for SSL, the labels we construct for each mini-batch during training vary across mini-batches.

---

> ### Author Response · Authors · 2024-11-21
> **Response to Weaknesses 4.2**
>
> From the second paragraph of **Section 2**, it can be inferred that both D-SSL (Discriminative SSL) and G-SSL (Generative SSL) can be viewed as aligning the positive sample in a pair with an anchor. Taking SimCLR as an example, it enforces the augmented sample that shares the same ancestor as the anchor to move closer to the anchor in the feature space, while other augmented samples move farther away.
>
> From this perspective, the anchor can be interpreted as a cluster center, representing a category label.

---

> ### Author Response · Authors · 2024-11-21
> **Response to Weaknesses 5.1**
>
> Thank you for pointing this out. Due to space limitations, in the **Appendix A.1** of the revised version, we have included the assumptions of the mixture of Gaussians, balanced labels, equal dimensions between spurious variables and images, and the binary classification model prior to presenting **Proposition 3.1**.

---

> ### Author Response · Authors · 2024-11-21
> **Response to Weaknesses 5.2**
>
> Thank you for pointing this out.  Based on **Section 2** and **Section 3.1**, under the new perspective, both D-SSL (Discriminative SSL) and G-SSL (Generative SSL) share a common learning objective: aligning the positive sample in a pair with its corresponding anchor. Thus, the learning objectives of D-SSL and G-SSL can be unified as maximizing $ p(x^{\rm{label}} | x^+) $. The difference lies in how they achieve $ p(x^{\rm{label}} | x^+) $: for example, contrastive learning uses a contrastive loss, while MAE employs the $ L_2 $-norm. In the revised version, we have formulated our statement in the beginning of **Theorem 3.4**.

---

> ### Author Response · Authors · 2024-11-21
> **Response to Weaknesses 6**
>
> Thank you for pointing these out. $ {\mathcal{L}}^{\rm PID} $ represents the loss computed on a dataset that satisfies the PID constraints. ${\perp}_{\rm PI}$ indeed denotes the independence condition satisfied for all PID.
>
> The definition of $ nu $ is provided in **Line 313** of the original submission, and the definition of $ mu $ is provided in **Line 333** of the original submission.

---

> ### Author Response · Authors · 2024-11-21
> **Response to Weaknesses 7**
>
> Thank you for pointing these out. The answer to this question can be effectively addressed by synthesizing the **Response to Weaknesses** 1 through 4.

---

> ### Author Response · Authors · 2024-11-21
> **Response to Weaknesses 8**
>
> Thank you for your comments. In **Table 1** and **Table 6** of the original submission, we reported the results on ImageNet-1K, where the models were trained for 1000 epochs.

---

> ### Author Response · Authors · 2024-11-21
> **Response to Weaknesses 9**
>
> Thank you for pointing this out. Due to space limitations, in the revised version, we have included this discussion in **Appendix D**.

---

> ### Author Response · Authors · 2024-11-21
> **Response to Question 1**
>
> Thank you for pointing this out. We made an error here; it should be "minimize."

---

> ### Comment · Reviewer_M58z · 2024-11-25
> **Reviewer's rebuttal response**
>
> Thank the authors for their thorough response, and my gratitude also goes to the other reviewers for their efforts. The rebuttal has fully addressed my concern regarding W1, W5, W6, W9. In particular, I appreciate the authors' discussion on the identifiability of spurious variables, which clearly states the condition where spurious variables are identifiable.
>
> However, my major concern remains over the relevance of the SCM model for OOD generalization of SSL (W2, W3) and a rigorous presentation of core concepts (W4). I agree that the proposed method has certified worst-case robustness by Theorem 3.4, but it has not shown, either theoretically or intuitively, why a non-spurious "alignment model" leads to a non-spurious generative model. As a counterexample raised in my initial review, if the augmentation procedure retains the style or background of the image, the classifier does not depend on the style to distinguish between anchor and augmentations because they share the same style, which indicates a non-spurious "alignment model". However, backgrounds are still learned by the generative model because they identify the features shared across x+ and x_label and discard others that are distinct.
>
> As another concept discussed by multiple reviewers, I would also like to point out that independence of s and X_label is totally different from conditional independence of s and X_label. The author claims conditional independence, while the SCM in Figure 3 actually indicates independence by d-separation. I suppose a clarification is necessary for the SCM used in this paper.
>
> Another concern over presentation is about the reformulation of OOD generalization as generalization on task distributions. I appreciate the authors' intuitive explanations in the rebuttal, but I do think the core concept should be rigorously formulated, because the current description is somewhat vague and dubious. As the authors claim that different tasks have different label types, it remains unresolved why generalization to unseen label types is possible. If comparing to classic supervise learning, generalization to unseen categorical attribute is impossible. In addition, it requires further discussion how the "task" relates to the SCM, the alignment model and the binary classification model in the following sections.
>
> Overall, I decide to maintain my current evaluation of this paper.

---

> ### Author Response · Authors · 2024-11-25
> **Response to "Reformulation of OOD generalization as generalization on task distributions"**
>
> Thank you for pointing out this issue. We organize our response into the following steps:
>
> ---
>
> ### **Step 1: First, we provide the formal definition of task distribution.**
>
> Without loss of generality, let us use a classification task as an example. We define $ {X_{tr}^a} = \\{({x_i^a}, {y_i^a}) \\}_{i = 1}^N $ as a training dataset, where $ x_i^a $ represents a sample, $ y_i^a $ represents the corresponding label, $ a $ denotes the dataset index, and $ N $ denotes the number of samples in the dataset. For a classification task, the goal is to learn a classifier $p^a(y_i^a | x_i^a) $, so that for any given sample $ x_i^a $, the corresponding label can be predicted.
>
> If $ N \to +\infty $, $ {X_{tr}^a} $ can be approximated as containing all the information necessary for the classification task and can thus be regarded as a complete dataset for a classification task. Simply put, the elements of a classification task include: the classifier and the dataset. We denote a task as $ (X_{tr}^a, p^a(y_i^a | x_i^a))$. Then, the discrete distribution of tasks can be expressed as $ \\{X_{tr}^a, p^a(y_i^a | x_i^a)\\}_{a = 1}^M $, where $ M $ represents the number of tasks.
>
> Furthermore, when $ a $ is different, the label space corresponding to $ y_i^a $ is also different. For example, when $ a=1 $, the label space is $ \\{Cat, Dog\\} $, and when $ a=2 $, the label space is $ \\{Plane, Train\\} $. If $ M \to +\infty $, $ \\{X_{tr}^a, p^a(y_i^a | x_i^a)\\}_{a = 1}^M $ can be regarded as a complete task distribution.
>
> ---
>
> ### **Step 2: Next, we reformulate SSL from the perspective of task distribution.**
>
> In **Section 2** and **Section 3.1-3.2** of  both original and revised submissions, we explain why a mini-batch in SSL can be viewed as a task. Simply, for a given mini-batch, it can be expressed as:
>
> $ X_{tr, a}^{aug} = \\{ x_{a}^i, x_{anchor, a}^{i} \\}_{i=1}^{2N} $,
>
> where $ N $ is the number of ancestor samples in the mini-batch, $ a $ is the index of the mini-batch, and $x_{anchor, a}^{i}$ is regarded as the label of the augmented sample. Meanwhile, the classifier to be learned for each mini-batch is modeled as $ p^a(x_{anchor, a}^{i} | x_{a}^i) $ (refer to lines 216-222).
>
> Notably, the classifiers for all tasks in SSL are learned using the same classifier, i.e., the classifiers for all tasks aim to learn $ p(x_{anchor, a}^{i} | x_{a}^i) $. For example, SimCLR models the classifier using a contrastive loss, while MAE models it using the $ L_2 $-norm. Therefore, whether D-SSL or G-SSL is used, as $ M \to +\infty $, $ \\{(X_{tr, a}^{aug}, p(x_{anchor, a}^{i} | x_{a}^i))\\}_{a=1}^{M} $ can be approximated as a task distribution, where $ M $ represents the number of tasks.
>
> ---
>
> ### **Step 3: Finally, we reformulate the OOD generalization of SSL as generalization on task distributions.**
>
> In traditional machine learning, given training data, the goal is to learn $ p(y | x) $. This can be understood as modeling the data distribution $ p(x, y) $ as $ p(x)p(y | x) $, where $ p(y | x) $ is learned from the training data and transferred to the test data distribution $ p(x) $. This approach assumes that the training and test data are identically and independently distributed, i.e., $ p(x_{train}) = p(x_{test}) = p(x) $, and $ p(x_{train}, y_{train}) = p(x_{test}, y_{test}) = p(x, y) $. Consequently, $ p(x)p^{train}(y | x) = p(x)p^{test}(y | x) $, leading to $ p^{train}(y | x) = p^{test}(y | x) $.
>
> By analogy, when each data sample is treated as a task, the corresponding learning objective becomes $ p(p^a(x_{anchor, a}^{i} | x_{a}^i) | X_{tr, a}^{aug}) $. This learning goal is similar to that in meta-learning [1-2], where the goal is to learn a function that can output the classifier for a given task dataset. Therefore, when the training data are drawn from a task distribution, the learning objective is to model the task distribution, i.e., to learn $ p(p^i(y | x) | \text{task } i) $, such that it applies to both training and test tasks. Since training and test tasks are different, from the perspective of the training tasks, the test tasks represent OOD scenarios. However, from the perspective of the task distribution, both training and test tasks belong to the same task distribution.
>
> Thus, from the viewpoint of traditional machine learning, SSL can be considered as training with mini-batches of size 1, where each training sample is a training task. One open problem is how to model $ p(p^i(y | x) | \text{task } i) $. Since we define the classifier $ p(x_{anchor, a}^{i} | x_{a}^i) $ for each SSL training task as identical, $ p(p^i(y | x) | \text{task } i) $ can be directly modeled as $p(x_{anchor, a}^{i} | x_{a}^i)$, which applies to any sample from any task.
>
> ---
>
> In conclusion, combining **Step 1-3**, we reformulate OOD generalization as generalization on task distributions.
>
> [1] Model-agnostic meta-learning for fast adaptation of deep networks;
>
> [2] The close relationship between contrastive learning and meta-learning;

---

> ### Author Response · Authors · 2024-11-25
> **Response to ''The relevance of the SCM model for OOD generalization of SSL''-----Part 1**
>
> Thank you for pointing out this issue again. **The reviewer has misunderstood a critical part of this paper: how to achieve good OOD generalization in SSL. This paper does not explain the effectiveness of PID from the perspective of eliminating spurious variables. Notablely, transferring Figure 1 to Figure 3 is similar to backdoor adjustment in causal inference. However, from backdoor adjustment pespective, it is straightforward to explain why PID can improve the OOD performance of D-SSL (Refer to literature 'Interventional Contrastive Learning with Meta Semantic Regularizer' for more details). However, for G-SSL, regardless of the relationship between $ s $ and $x^{\rm label}$, G-SSL inherently requires encoding background semantics. Thus, explaining the improvement of OOD performance of G-SSL using the backdoor adjustment is incorrect. Theorem 3.4 is provided to explain why PID can improve the OOD performance of both D-SSL and G-SSL simultaneously.**
>
> **The validity of PID in this paper is justified from the worst-case perspective, as stated in Theorem 3.4.**
>
> To make it easier for the reviewers and readers to understand, we further clarify the logical structure and viewpoints of this paper in the following steps:
>
> ---
>
> ### **Step 1: Reformulate SSL from the perspective of task distribution**
>
> From the perspective of task distribution, SSL can be understood as learning a distribution of tasks. Each task during training is a classification task: for G-SSL, the classifier is modeled using the $ L_2 $-norm, while for D-SSL, the classifier is modeled using contrastive loss. It is crucial to emphasize that we unify the concepts of alignment, classifier, and loss function, as they are essentially the same in our formulation. For further details, please refer to **Response to "Reformulation of OOD generalization as generalization on task distributions"**.
>
> ---
>
> ### **Step 2: Model the learning process of each task using a fuzzy SCM**
>
> In our new understanding, we model the learning process of each task using a fuzzy SCM. It is considered "fuzzy" because the relationship between $ s $ and $ x^{\rm label} $ is unclear, as shown in **Figure 1**. We elaborate on this in **Lines 153-176** of the revised submission, explaining that the relationship between $ s $ and $ x^{\rm label} $ varies across tasks and cannot be captured using a single SCM.
>
> ---
>
> ### **Step 3: Explain how spurious variables impact OOD generalization in SSL**
>
> In **Lines 153-181** of the revised submission, we explain that under our new understanding, current SSL methods may learn spurious variables, which negatively impact OOD generalization performance. Specifically, spurious variables affect OOD generalization as follows:
>
> - Spurious variables make it challenging to learn each specific task properly.
> - This, in turn, hinders the modeling of the task distribution.
> - Consequently, SSL performs poorly on test tasks, where test tasks differ from training tasks.
> - This ultimately undermines OOD generalization.
>
> ---
>
> ### **Step 4: Core argument — improving SSL's OOD performance despite spurious variables**
>
> At this point, the core argument of this paper emerges. Through **Theorem 3.4**, we demonstrate that even in the presence of spurious variables, it is possible to propose a method to enhance OOD generalization.
>
> From **Assumption 3.3**, the optimal $ f^* $ should be $ F_{x^{\rm label}} $. However, without additional constraints, obtaining the optimal $ f^* $ is challenging—**a key concern raised by the reviewer.**
>
> **Theorem 3.4** implies that when $\mathcal{D} $ is sufficiently large and diverse, an optimal $ f^* $ trained on one distribution will perform worse than random guessing in some other environments. Under such conditions, no other $ f $ obtained from training on any distribution can achieve better worst-case OOD performance than the PID.
>
> **In other words, even in the presence of spurious variables, e.g. G-SSL,  our proposed approach can improve the OOD performance of SSL.**
>
> ---
>
> ### **Step 5: Why is the worst-case scenario critical for improving SSL's OOD performance?**
>
> **This is a key insight into how our approach improves SSL's OOD performance.**
> During training, we minimize the loss in the worst-case scenario: $
> {\max _{e \in \mathcal{D}}} {\mathcal{L}}^e({p_f}(x^{\rm{label}} | x^+))
> $. For any $ f $, the worst-case loss $
> {\max _{e \in \mathcal{D}}} {\mathcal{L}}^e({p_f}(x^{\rm{label}} | x^+))
> $ is always greater than or equal to the loss in any specific environment $e$: $
> {\mathcal{L}}^e({p_f}(x^{\rm{label}} | x^+)).
> $
>
> If we learn an $ f $ that minimizes the worst-case loss $
> {\max _{e \in \mathcal{D}}} {\mathcal{L}}^e({p_f}(x^{\rm{label}} | x^+))
> $, then $ f $ naturally minimizes $
> {\mathcal{L}}^e({p_f}(x^{\rm{label}} | x^+))
> $ for all $ e $ in $ \mathcal{D} $. This ensures robust performance across all scenarios, making the worst-case optimization strategy particularly effective for enhancing OOD generalization.

---

> ### Author Response · Authors · 2024-11-25
> **Response to ''The relevance of the SCM model for OOD generalization of SSL''-----Part 2**
>
> ### **Step 6: The proposed approach — achieving PID**
>
> Our approach achieves PID based on SCM. Why does **Algorithm 1** achieve PID? A high-level explanation is as follows:
>
> 1) From **Definition 4.4**, it follows that if $ ba(s) $ can be identified, then $ s $ and $ x^{\rm label} $ are conditionally independent given $ ba(s) $.
>
> 2) In this paper, $ ba(s) $ is implemented as described in Equation (5) in the main text. The key challenge lies in obtaining $ s $. In our response to **Reviewer 1's first weakness**, we explained the identifiability of $ s $, as well as how each label is modeled using a distribution for $ s $. During implementation, we sample from this distribution to generate a series of discrete vectors that approximate $ s $ associated with a specific label.
>
> 3) From **Equation (2)** in the main text, we have: $
>    p(x^+, x^{\rm{label}}, s) = p(x^+ | x^{\rm{label}}, s) p(x^{\rm{label}}) p(s | x^{\rm{label}})
>    $. If we select sample pairs for a mini-batch such that all pairs share the same $ ba(s) $, the resulting mini-batch can be considered as constructed under the same $ ba(s) $. In other words, the samples in the mini-batch are conditioned on $ ba(s) $. Combined with the argument in **Point 1**, we have: $
>    p(x^+, x^{\rm{label}}, s) = p(x^+ | x^{\rm{label}}, s) p(x^{\rm{label}}) p(s)
>    $, which ensures that the mini-batch satisfies PID.
>
> The key to achieving PID is ensuring that all sampled examples in the mini-batch share consistent \( ba(s) \), i.e., the background information is the same. This allows SSL to focus on foreground information while disregarding background information.
>
> ---
>
> ### **Addressing the Reviewer's Questions**
>
> Based on the above explanation, the reviewer's concerns, including:
>
> - "Why does a non-spurious alignment model lead to a non-spurious generative model?"
> - "The counterexample raised by the reviewer."
> - "Backgrounds are still learned by the generative model because they identify the features shared across \( x^+ \) and \( x_{\rm label} \) and discard others that are distinct."
>
> can all be directly resolved. Our method does not improve SSL's OOD performance by eliminating spurious variables but rather by leveraging a worst-case scenario strategy to enhance OOD performance. Regarding the SCM, it serves two purposes: to explain the limitations of existing SSL methods and to justify the rationale behind our proposed approach.

---

> ### Author Response · Authors · 2024-11-25
> **Response to "Independence of $s$ and $X^{label}$"**
>
> Thank you for pointing out this issue again. **Assumption 3.3** illustrates that, whether $ e \in \mathcal{D} $ or $ e \in \text{PID} $, $ x^+ $ is generated under the control of two variables, $ s $ and $ x^{\rm label} $. Therefore, given $ x^+ $, $ s $ and $ x^{\rm label} $ are conditionally independent, regardless of any correlation between them.
>
> ---
>
> In **Figure 3**, our PID is indeed based on the independence of $ s $ and $ x^{\rm label} $. Regarding the method we use to achieve this independence, it does not rely on techniques related to d-separation from causal inference. Instead, we achieve PID through **Algorithm 1**, which is closely related to the average treatment effect estimation.
>
> Why does **Algorithm 1** achieve PID? The high-level explanation is as follows:
>
> ---
>
> 1) From **Definition 4.4**, it follows that if $ ba(s) $ can be identified, then $ s $ and $ x^{\rm label} $ are conditionally independent given $ ba(s) $.
>
> 2) In this paper, $ ba(s) $ is implemented as described in Equation (5) in the main text. The key challenge lies in obtaining $ s $. In our response to **Reviewer 1's first weakness**, we explained the identifiability of $ s $, as well as how each label is modeled using a distribution for $ s $. During implementation, we sample from this distribution to generate a series of discrete vectors that approximate $ s $ associated with a specific label.
>
> 3) From **Equation (2)** in the main text, we have: $
>    p(x^+, x^{\rm{label}}, s) = p(x^+ | x^{\rm{label}}, s) p(x^{\rm{label}}) p(s | x^{\rm{label}})
>    $. If we select sample pairs for a mini-batch such that all pairs share the same $ ba(s) $, the resulting mini-batch can be considered as constructed under the same $ ba(s) $. In other words, the samples in the mini-batch are conditioned on $ ba(s) $. Combined with the argument in **Point 1**, we have: $
>    p(x^+, x^{\rm{label}}, s) = p(x^+ | x^{\rm{label}}, s) p(x^{\rm{label}}) p(s)
>    $, which ensures that the mini-batch satisfies PID.
>
> ---
>
> The key to achieving PID is ensuring that all sampled examples in the mini-batch have consistent $ ba(s) $, i.e., the background information is the same. This enables SSL training to focus on the foreground information while disregarding the background information.

---

> ### Comment · Reviewer_M58z · 2024-12-03
> **Post-rebuttal comment of the reviewer.**
>
> Thank you to the author for their detailed and thoughtful discussion. Initially, I mentioned the major weakness concerning the relevance of the proposed method to OOD generalization for SSL, as well as major weaknesses in the writing, particularly in the definition of tasks.  I also raised several minor points.
>
> The author has effectively addressed the major writing weaknesses by clearly formulating the definition of tasks in the rebuttal and has resolved all minor points through clarification. Therefore, I have adjusted my score from 3 to 5.
>
> However, my remaining concern is about the relevance of the proposed method for OOD generalization in SSL. While I appreciate the author explicitly acknowledging that the paper does not address the elimination of spurious variables in SSL learning, I disagree with their claim that the worst-case analysis of the alignment model presented in Theorem 3.4 is directly relevant to OOD generalization for SSL. I think the analysis would be more appropriately to be conducted on downstream models built upon SSL learning, which is not addressed in this paper.

---

> ### Author Response · Authors · 2024-12-03
> **Response to "OOD generalization in SSL"**
>
> Thank you for pointing out this issue again. The reviewer may have misunderstood what we meant.
>
> ---
>
> **What we need to clarify again is that:**
>
> 1) This paper focus on how to **deal with** the spurious variables in SSL learning, this **does not mean elimination**.
>
> 2) The proposed PID can be directly adapted to D-SSL and G-SSL.
>
> 3) **The most importantly is that**:  For D-SSL, the proposed PID can be understood as **elimination of spurious variables** in SSL learning. For G-SSL, the proposed PID **can not be understood** as **elimination of spurious variables** in SSL learning. Thus, we propose **Theorem 3.4**.
>
> **Theorem 3.4** can be viewed as **a new perspective** to **explain the effectiveness** of proposed PID in **deal with the spurious variables**, this perspective is **different and not original from elimination**.
>
> **Theorem 3.4** can be see as a **Universal Interpretation Framework** for both D-SSL and G-SSL. **Elimination-based perspective is only suitable for D-SSL**.
>
> ---
>
> We do not agree that **Theorem 3.4** would be more appropriately to be conducted on downstream models built upon SSL learning. We present the following reasons:
>
> **OOD generalization in SSL**:
>
> 1) The proposed PID is **a mini-batch sampling method** and suitable for all related D-SSL and G-SSL. Also, we have conducted many OOD-related experiments to demonstrate the effectiveness of the proposed PID, e.g., semi-supervised task, transfer task, few-shot task, and classification task that related to OOD dataset (PACS, OfficeHome, and ColoredMNIST dataset).
>
> 2) As shown in **Response to Reformulation of OOD generalization as generalization on task distributions**, we can obtain that SSL can be viewed as learning based on tasks. In other word, the key idea of this perspective is that **modeling task distribution**, the training samples is a serious of training tasks, and the test samples is also a series of training tasks. **The training tasks and the test task is different**, **this is the key concept of the OOD generalization**.
>
> 3) **Theorem 3.4** is built on the new understanding of SSL shown in "2)". Under the new understanding of SSL, **Theorem 3.4** propose a new perspective, e.g., the worst-case analysis, to demonstrate the effectiveness of PID in both D-SSL and G-SSL.
>
> **Further explanation on Why "worst-case analysis" is good for OOD generalization in SSL**?
>
> 1) From **Step 5** in **Response to The relevance of the SCM model for OOD generalization of SSL-----Part 1**, we have demonstrated that **the worst-case** can help us to better learning $p(x_{anchor, a}^{i} | x_{a}^i)$. A better $p(x_{anchor, a}^{i} | x_{a}^i)$ can **achieve a less empirical risk in the training distribution**.  As shown in **Step 3** of **Response to Reformulation of OOD generalization as generalization on task distributions**, from the viewpoint of traditional machine learning, SSL can be considered as training with mini-batches of size 1, where each training sample is a training task. Then, from PAC theory, we can obtain that minimizing the Empirical Risk **can lead to minimize Expected Risk**, and the **Expected Risk** is calculated based on a series of test tasks. Thus,  **the worst-case** can improve the OOD generalization in SSL.

---

### Meta-Review · Area_Chair_P4F7 · 2024-12-19

**Metareview:**

Summary: This paper explores self-supervised learning (SSL) from a causal perspective, proposing a novel training batch sampling strategy to mitigate spurious correlations between images and non-semantic features, such as backgrounds and styles. The approach leverages Structural Causal Models (SCMs) to model augmentation processes in both generative and discriminative SSL frameworks. By treating mini-batches as distinct environments in out-of-distribution (OOD) generalization problems, the authors introduce a method that rebalances training batches using a Post-Intervention Distribution (PID) modeled by Variational Autoencoders (VAEs). This ensures that sampled images are decorrelated from spurious features, aligning with invariant causal structures across environments. Experiments demonstrate improved generalization and performance across various SSL methods.

Strengths:
1. The proposed rebalancing technique is with wide applicability and theoretical guarantee.
2. Experiments cover multiple scopes and scenarios.

Weaknesses (after the rebuttal):
1. Reviewers are concerned about the relevance of the SCM model for OOD generalization of SSL.
2. Some concepts and statements are not well defined and formulated.
3. Reviewers have concerns on the presentation about the reformulation of OOD generalization as generalization on task distributions.
4. Reviewers are confused about how Theorem 3.4 implies that "when is sufficiently large and diverse, an optimal trained on one distribution will perform worse than random guessing in some other environment."?

The paper is on the broadline, while no reviewer is willing to champion the paper. Given the remaining concerns after rebuttal, AC would recommend rejection and encourage the authors to revise the paper to mitigate confusion in the next version of this paper.

**Additional Comments On Reviewer Discussion:**

All reviewers rate the paper with a broadline score. Two vote for marginal above the bar, while another votes for marginal below the bar.

After the rebuttal, Reviewer fPhX has concerns on Question 2: Authors' response seems to emphasize that PID achieves the optimal worst-case OOD performance, but it doesn't directly resolve the reviewer's confusion.

Most of Reviewer M58z's concerns have been addressed in the rebuttal, and the reviewer has raised his/her score from 3 to 5. However, one concern remains about the relevance of the proposed method for OOD generalization in SSL. In particular, the reviewer disagrees with the claim that the worst-case analysis of the alignment model presented in Theorem 3.4 is directly relevant to OOD generalization for SSL. The reviewer thinks the analysis would be more appropriate to be conducted on downstream models built upon SSL learning, which is not addressed in this paper. The SAC has also checked the reviews and rebuttals and agreed wth the AC for the final decision.

---

### Decision · Program_Chairs · 2025-01-22

Reject